# Gpc3 selectively suppresses subcutaneous adipogenesis in diet-induced obesity

Yan Li[1,2], Ming Tao[3], Carlos F. Ibáñez[1,4,5,6,7]*, Meng Xie [5,6,8,9]*

1 Chinese Institute for Brain Research, Zhongguancun Life Science Park, Beijing, China, 2 Academy for Advanced Interdisciplinary Studies, Peking University, Beijing, China, 3 Department of General Surgery, Peking University Third Hospital, Beijing, China, 4 School of Life Sciences, Peking University, Beijing, China, 5 Peking-Tsinghua Center for Life Sciences, Beijing, China, 6 PKU-IDG/McGovern Institute for Brain Research, Beijing, China, 7 Department of Neuroscience, Karolinska Institute, Stockholm, Sweden, 8 School of Psychological and Cognitive Sciences, Peking University, Beijing Key Laboratory of Behavior and Mental Health, Beijing, China, 9 Biosciences and Nutrition Unit, Department of Medicine Huddinge, Karolinska Institute, Huddinge, Sweden

* carlos.ibanez@pku.edu.cn (CFI); meng.xie@pku.edu.cn (MX)

## Abstract

Subcutaneous and visceral adipose depots employ distinct expansion strategies in response to dietary cues, yet the molecular regulators underlying these depot-specific adaptations remain poorly understood. Through integrated proteomic profiling of human subcutaneous and visceral adipose tissues from paired obese/non-obese donors and temporal transcriptomic analysis of mouse adipose stem and progenitor cells (ASPCs) during dietary transitions, we identified Glypican 3 (Gpc3) as an obesity-responsive gene exhibiting reciprocal expression patterns between depots. ASPC-specific Gpc3 deletion in mice amplified high-fat diet-induced weight and fat mass gain, with a selective enhancement of expansion in inguinal white adipose tissue (WAT) without affecting epididymal WAT. Mechanistically, Gpc3 loss biased ASPC fate toward adipogenesis over proliferation through depot-specific modulation of canonical Wnt signaling. These findings establish Gpc3 as a regulator for regional adipose plasticity, offering a molecular target for reprogramming pathological fat distribution in obesity and related metabolic disorders.

## Introduction

White adipose tissue (WAT) serves as the major energy reservoir in mammals, storing excess energy in unilocular adipocytes under conditions of prolonged energy surplus such as chronic high-fat diet (HFD) feeding. In addition to adipocytes, WAT contains a heterogeneous stromal vascular fraction (SVF) composed of adipose stem and progenitor cells (ASPCs), immune cells, and vascular cells. WAT expands through two major mechanisms: hypertrophic growth of existing adipocytes and hyperplastic expansion followed by de novo adipogenesis from ASPCs. Anatomically, WAT is organized into subcutaneous and visceral depots, which exhibit fundamental

**Data availability statement:** All transcriptomic data generated in this study were deposited in the Gene Expression Omnibus (GSE298345) https://www.ncbi.nlm.nih.gov/geo/query/acc.cgi?acc=GSE298345. All other underlying data can be found within the manuscript file and supplementary material. All codes used in this study are available at Zenodo (DOI: https://doi.org/10.5281/zenodo.18693311) and GitHub: (https://github.com/YanLi0519/Gpc3_project).

**Funding:** This work was supported by research grants to CFI from Peking University, Chinese Institute for Brain Research, Beijing, Swedish Cancer Society (Cancerfonden, contract no. 222135Pj01H) and Swedish Research Council (Vetenskapsrådet, contract no. 2020-01923_3); and a startup grant to MX from Swedish Research Council (Vetenskapsrådet, contract no. 2021-01805). The funders had no role in study design, data collection and analysis, decision to publish, or preparation of the manuscript.

**Competing interests:** The authors have declared that no competing interests exist.

**Abbreviations**: ASAT, abdominal subcutaneous; ASPCs, adipose stem and progenitor cells; CD, chow diet; DEG, differentially expressed gene; DIA, data-independent acquisition; DIV, days in vitro; ECM, extracellular matrix; FSC, forward scatter; GO, Gene Ontology; Gpc3, Glypican 3; HE, hematoxylin and eosin; HFD, high-fat diet; IACUC, Institutional Animal Care and Use Committees; PFA, paraformaldehyde; RNP, ribonucleoprotein; RT, room temperature; rWnt3a, recombinant Wnt3a; scRNA-seq, single-cell RNA sequencing; ssGSEA, Single-sample Gene Set Enrichment Analysis; SVF, stromal vascular fraction; OVAT, omental visceral adipose tissue; WAT, white adipose tissue.

differences in gene expression profiles [1–3], developmental origins [4,5], and secretory functions [6]. While depot-specific microenvironmental cues have been implicated in driving adipocyte hyperplasia during obesity [7], the contribution of cell-autonomous mechanisms in ASPCs to depot-specific adipose plasticity remains incompletely understood.

Advances in single-cell transcriptomics have greatly improved our understanding of adipose tissue heterogeneity, revealing multiple progenitor subpopulations with distinct adipogenic, fibrogenic, and immunomodulatory functions [8–13]. In addition, recent multi-omics studies have provided deeper insights into the diverse states and functions of ASPCs in different adipose depots. Integrated transcriptomic and proteomic profiling of ASPCs from inguinal (iWAT) and gonadal WAT revealed thousands of depot- and sex-dependent protein expression differences, highlighting substantial intrinsic depot heterogeneity [14]. In human WAT, ASPC subpopulations displayed divergent lipid metabolism and anatomical distribution across subcutaneous and omental depots, with alterations associated with type 2 diabetes [15]. Notably, ASPCs from iWAT demonstrate significantly greater in vitro adipogenic capacity than those from gonadal WAT [16], reflecting intrinsic molecular and functional differences between subcutaneous and visceral depots [17]. However, most of these studies focused on static metabolic states and lacked integrative analyses across human and murine datasets. As a result, how subcutaneous and visceral depots are dynamically remodeled in response to metabolic changes remains unclear, limiting our understanding of their distinct capacities for expansion, adaptation, and metabolic protection.

To investigate how different adipose depots respond to metabolic changes, we performed proteomics analysis on paired subcutaneous and visceral adipose samples from obese and lean human donors, as well as single-cell RNA sequencing (scRNA-seq) on iWAT and epididymal WAT (eWAT) from mice that experienced switches between HFD and chow diet (CD). These approaches identified Glypican 3 (Gpc3) as an evolutionarily conserved and depot-specific regulatory switch that differentially responds to changing metabolic demands. Gpc3 is a membrane-anchored heparan sulfate proteoglycan that modulates cellular growth, differentiation and migration through regulation of Wnt, Hedgehog, and growth factor signaling pathways [18,19]. While best characterized as a clinical biomarker for hepatocellular carcinoma, Gpc3 is also enriched in fibro-inflammatory progenitors [9] and brown adipocyte progenitors [20] of adipose tissue. Outside liver cancer, however, the physiological roles of Gpc3 in other tissues, particularly adipose depots, have not been delineated. Here, we show that Gpc3 specifically promotes iWAT ASPC hyperplasia during HFD-feeding through canonical Wnt signaling.

## Results

### Human subcutaneous and visceral adipose tissue exhibit distinct proteomic profiles irrespective of obesity status

To investigate depot-specific proteomic differences in humans, we performed quantitative proteomic analysis using data-independent acquisition (DIA) on paired

abdominal subcutaneous (ASAT) and omental visceral adipose tissue (OVAT) samples from four non-obese (BMI < 30, Subject 1–4) and four obese (BMI > 35, Subject 5–8) individuals (Fig 1A) (S1 Table). Strikingly, pairwise comparison of protein abundance between the two depots within each subject revealed a two- to 10-fold higher percentage of enriched proteins in OVAT across all individuals (Fig 1B), indicating profound biological and functional divergence between these depots, independent of obesity status. Intersection analysis of ASAT- and OVAT-enriched proteins across the four sub- jects of the obese and non-obese groups identified 144 (ASAT non-obese), 138 (ASAT obese), 2,823 (OVAT non-obese), and 3,115 (OVAT obese) proteins, respectively (Fig 1C). The top 20 enriched proteins for each group are listed in Fig 1D. Pathway enrichment analyses demonstrated depot-specific molecular programs that were preserved between the obese and non-obese groups (Fig 1E). ASAT-enriched proteins were associated with reactive oxygen species metabolism, cellular detoxification, and gas transport, suggesting a state of high metabolic activity that generates oxidative stress in this tissue. In contrast, OVAT-enriched proteins from both lean and obese individuals showed enrichment in transcriptional and post-transcriptional regulatory processes such as RNA splicing, ribonucleoprotein (RNP) complex biogenesis, and protein-RNA complex organization, suggesting that it maintains a biosynthetically active environment, supporting dynamic RNA metabolism and efficient protein synthesis. Notably, these depot-specific pathway signatures were maintained regardless of obesity status (Fig 1E), suggesting that the fundamental identity and function of adipose tissue is primarily defined by its anatomical location, with obesity exacerbating rather than reprogramming these innate functions.

**Mouse subcutaneous and visceral ASPCs exhibit depot-specific transcriptional plasticity in response to dietary transitions**

The profound proteomic differences between human ASAT and OVAT prompted us to investigate the cellular origins of this depot-specific identity. We reasoned that ASPCs, as the reservoir for new adipocytes, are likely key determinants of these distinct tissue phenotypes. Therefore, we turned to a murine model to enable a high-resolution analysis of ASPC tran- scriptomic plasticity in response to metabolic challenge. To investigate the transcriptomic plasticity of subcutaneous and visceral ASPCs in response to transitions between CD and HFD, we set up the following five groups of feeding paradigms in C57BL/6J male mice: mice switched from CD to HFD at 6 weeks of age for either 12 (12wHFD) or 18 weeks (18wHFD); control groups maintained on CD for equivalent durations (12wCD, 18wCD); and a reversal group subjected to 12 weeks HFD followed by 6 weeks CD (12wHFD-6wCD) (Fig 2A). At the end of this schedule (i.e., 24 weeks of age), SVFs from iWAT and eWAT were isolated for scRNA-seq analysis. A total of 86,134 cells were identified after rigorous filtering of the 10 integrated datasets (S1A Fig). Unsupervised clustering identified expected cell populations, including ASPCs, immune lineages (macrophage, B cell, T cell, dendritic cell, neutrophil, natural killer cell, mast cell), and vascular components (peri- cyte, endothelial cell, and mesothelial cell) (S1B and S1C Fig). Further clustering of the ASPCs revealed *Dpp4+&Pi16+* progenitors (ASPC1) and *Pdgfrb+&Icam1+* preadipocytes (ASPC2) (Figs 2B and S1D). This dietary shift from CD to HFD induced a proportional shift from ASPC1 to ASPC2 at both 12 and 18 weeks. The effect was most pronounced in eWAT, with iWAT showing a less obvious trend, and was reversed upon returning to a CD (Figs 2C and S2). This suggests that the balance between progenitor and preadipocyte subpopulations is a dynamic equilibrium highly sensitive to and revers- ible by environmental nutritional signals. At the depot level, eWAT had a higher proportion of ASPC2 than iWAT across all dietary conditions (Fig 2C), suggesting a depot-specific difference in ASPC subpopulation composition.

To elucidate the depot-specific molecular underpinnings of adipose tissue plasticity in response to dietary transitions, we first performed differentially expressed gene (DEG) analysis on ASPCs of iWAT and eWAT between CD- and HFD-fed cohorts at the 12- (12wCD versus 12wHFD) and 18-week time points (18wCD versus 18wHFD). Then, we performed an intersection analysis of the upregulated and downregulated gene sets from the 12wHFD and 18wHFD groups to identify conserved transcriptional changes that persisted across both feeding durations. Using stringent thresholds (detection in ≥25% of cells, |log2FC| > 0.25, adjusted $p < 0.05$), we identified 87 HFD-responsive genes in iWAT ASPCs (56 upreg- ulated and 31 downregulated) (Fig 2D). Consistent with the human proteomic analysis (Fig 1B and 1C), eWAT ASPCs

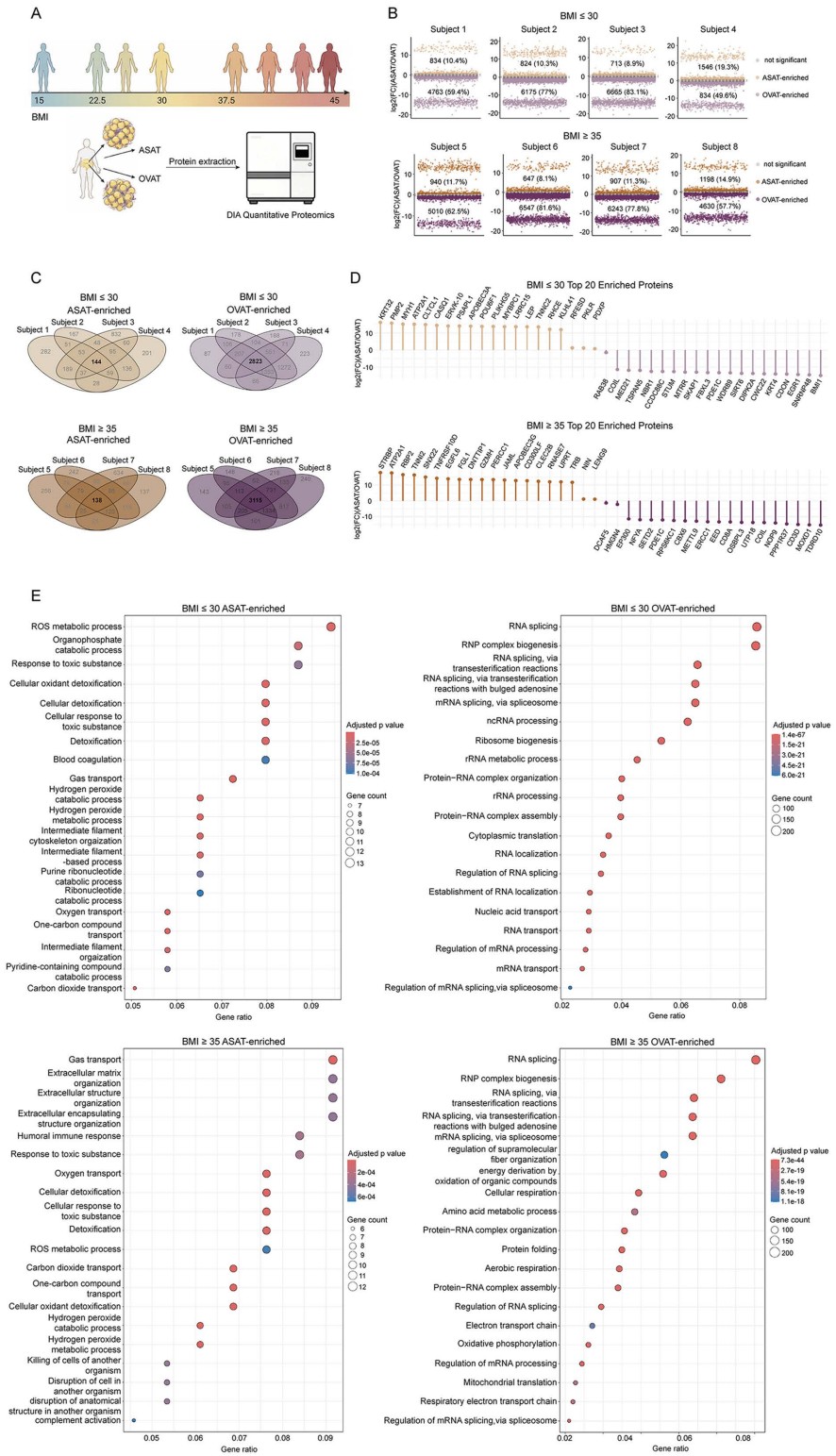

**Fig 1. Human subcutaneous and visceral adipose depots maintain distinct proteomic signatures across metabolic states. (A)** Schematic overview of the experimental pipeline for DIA quantitative proteomic analysis of human ASAT and OVAT. Created by hand. **(B)** Scatter plots comparing protein abundances in ASAT vs. OVAT for each individual. *n* = 4 human subjects per group. Points above/below the dashed lines denote proteins

enriched in ASAT or OVAT. Non-obese and obese groups were denoted with weak and strong color intensities, respectively. **(C)** Venn diagrams identify overlapping enriched proteins present in all 4 obese or non-obese subjects. **(D)** Lollipop plots rank the top 20 depot-enriched proteins from **(C)**. **(E)** GO analysis (biological process) of ASAT- and OVAT-enriched proteins. Top 20 enriched pathways were shown for each group.

exhibited a more extensive transcriptional response with 258 upregulated and 282 downregulated genes, indicating greater metabolic plasticity in the visceral depot. Importantly, both the expression magnitude (Fig 2E) and cellular prevalence (Fig 2F) of these HFD-responsive genes reverted toward baseline levels in the 12wHFD-6wCD switch group of both depots, demonstrating the dynamic adaptability of ASPCs to nutritional cues. Pathway enrichment analysis among HFD-responsive genes revealed distinct signaling pathways between iWAT and eWAT (Fig 2G), suggesting depot-specific transcriptional plasticity in response to dietary transitions.

## Gpc3 displays reciprocal expression dynamics in subcutaneous versus visceral WAT during metabolic adaptation in mice and humans

Among all HFD-responsive genes analyzed, *Gpc3* emerged as the only gene that displayed reciprocal expression dynamics between iWAT and eWAT in response to HFD and CD transitions. In iWAT, both *Gpc3* expression levels and the proportion of *Gpc3*-expressing ASPCs increased under HFD and decreased upon reversion to CD (Fig 3A and 3B). Conversely, eWAT demonstrated the complete inverse response pattern (Fig 3A and 3B). This depot-specific regulation was further confirmed at the mRNA level in whole-tissue analysis (Fig 3C) and the protein level in SVF (Fig 3D). Consistent with the murine data, DIA quantitative proteomic analysis of paired ASAT and OVAT from obese and lean donors revealed similar differences in GPC3 protein levels (Fig 3E). Notably, western blot analysis detected GPC3 protein in ASAT from two of the obese subjects (Subject 5 and 7) from the proteomics analysis as well as two additional obese subjects (Subject 9 and 10), with no expression observed in OVAT (Fig 3F). In addition, we found a modest positive association between GPC3 protein expression and BMI in the ASAT of the obese donor group (Fig 3G). A similar trend between *Gpc3* mRNA level and body weight was seen in the iWAT of HFD-fed mice, whereas their eWAT exhibited a negative correlation (Fig 3H). Moreover, analysis of published single-nucleus RNA sequencing (snRNA-seq) datasets [12] from human subcutaneous and visceral adipose tissue further corroborated that *GPC3* expression was predominantly localized to ASPCs (Fig 3I). Importantly, both fat depots exhibited reciprocal *GPC3* expression patterns between lean (BMI 20–30) and obese (BMI 30–40) donors, mirroring the responses observed in mouse dietary interventions (Fig 3I). Collectively, these findings support an evolutionarily conserved, depot-specific role for Gpc3 in adaptative responses to metabolic challenges.

## ASPC-specific Gpc3 deletion selectively promotes HFD-induced iWAT expansion

To investigate the cell-autonomous role of Gpc3 in ASPCs, we obtained a mouse line harboring a conditional knockout allele of the *Gpc3* gene (located on the X chromosome) with loxP sites flanking exon 3. Crossing these mice with *PdgfraCre* mice enabled the deletion of *Gpc3* specifically in ASPCs. qPCR analysis confirmed significant reductions in *Gpc3* mRNA levels in both iWAT and eWAT of *PdgfraCre*;*Gpc3^flox^* mice (mutant) compared to the *Gpc3^flox^* littermates (control) (Fig 4A). Consistently, ASPCs isolated from both depots showed comparable knockdown efficiency, with near-complete loss of *Gpc3* expression (Fig 4B). Next, we subjected 6-week-old control and mutant mice to an 18-week HFD regimen. Mutant mice showed significantly greater weight gain and fat mass accumulation compared to both control and *PdgfraCre* mice (Fig 4C). At the tissue level, the wet weight of iWAT but not eWAT was significantly increased in the mutant mice (Fig 4D). Consistent with this observation, iWAT adipocytes in mutant mice were significantly larger in size, whereas eWAT adipocyte size remained comparable between groups (Fig 4E). Notably, control and mutant mice showed no differences in glucose tolerance, insulin sensitivity, respiration, energy expenditure, food intake, and ambulation (Figs 4F, 4G, and S3A), suggesting that the mutation primarily affects local adipose tissue expansion rather than systemic metabolic regulation.

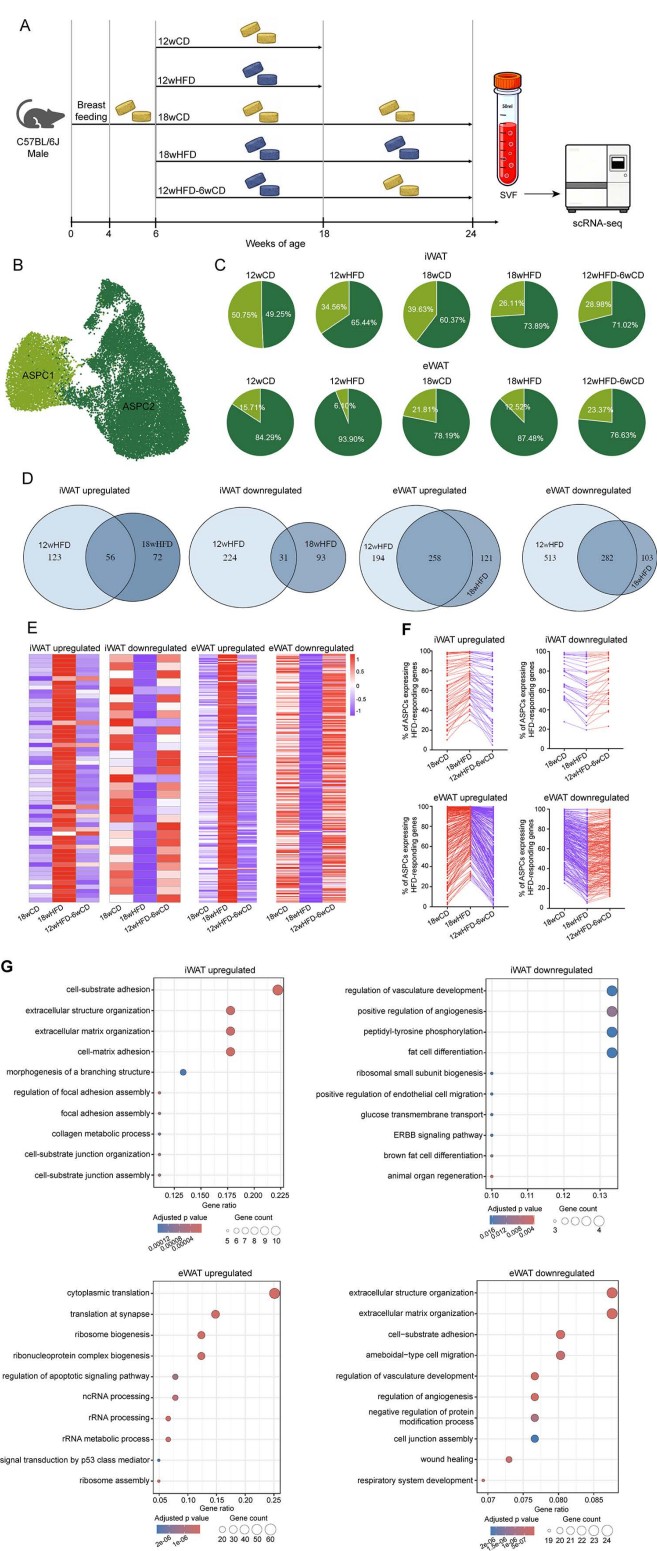

**Fig 2. Mouse ASPCs of subcutaneous and visceral WAT show distinct transcriptomic plasticity in response to dietary switches. (A)** Schematic illustration of the feeding paradigms for the five groups of C57BL/6J male mice. CD, yellow pellet; HFD, blue pellet. Created by hand. **(B)** Uniform Manifold Approximation and Projection (UMAP) of integrated ASPC populations from the 10 scRNA-seq datasets. **(C)** Pie charts quantify the proportion

of ASPC1 and ASPC2 in iWAT and eWAT. **(D)** Venn diagrams identify overlapping HFD-responsive genes in ASPCs from 12wHFD and 18wHFD groups. **(E)** Heatmaps show changes in HFD-responsive gene expression level in the 18wCD, 18wHFD, and 12wHFD-6wCD groups. Z-score was used for presentation. **(F)** Line plots show changes in the percentage of ASPCs expressing individual HFD-responsive genes. **(G)** GO analysis of the HFD-responsive genes in iWAT and eWAT. Top 10 enriched pathways were shown for each group.

To identify molecular pathways driving depot-specific adipose remodeling, we performed quantitative DIA proteomics on iWAT and eWAT from HFD-fed control and mutant mice. We found that most of the pathways that are known to regulate adipose tissue expansion, including lipid uptake and storage, lipogenesis, adipogenesis, inflammation, extracellular matrix (ECM) remodeling, autophagy and lysosomal function, and hormonal regulation were upregulated in the iWAT of mutant mice (Fig 4H). In contrast, these pathways were either unchanged or downregulated in eWAT (Fig 4H), highlighting a fundamental depot-specific role for Gpc3 that promotes metabolically favorable remodeling in iWAT. Together, these results demonstrate that *Gpc3* deletion in ASPCs drives depot-specific adipose expansion under HFD-feeding by selectively affecting iWAT homeostasis, possibly through coordinated upregulation of lipid metabolic, inflammatory, and remodeling pathways. snRNA-seq of iWAT from HFD-fed mice confirmed the efficacy of the genetic model, revealing a 3-fold reduction in the proportion of cells expressing *Gpc3* (predominantly ASPCs) in mutants compared to controls (Fig 4I). While *Gpc3* deletion did not drastically alter the proportional distribution of the two major ASPC subpopulations (Fig 4J), it profoundly altered their transcriptional programs as reflected by a shift in the top 5 enriched pathways within both populations (Fig 4K). Specifically, ASPC1 transitioned from a thermogenic profile in controls to an immunomodulatory state in mutants, whereas ASPC2 shifted its enrichment from ECM organization to pathways governing translational regulation (Fig 4K).

Of note, *Gpc3* mRNA levels showed a decreasing trend in the brain, but not in the pancreas, of mutant mice (S3B Fig). To determine if brain Gpc3 contributes to the phenotype, we generated *Camk2aCre*;*Gpc3^flox^* mice to delete *Gpc3* specifically in forebrain neurons. On the same HFD regimen, these mice showed no differences in body weight or fat mass compared to *Gpc3^flox^* controls (S3C Fig), indicating that brain Gpc3 does not play a major role in the observed obesity. Consistent with its location on the X chromosome, qPCR analysis revealed that *Gpc3* mRNA abundance was significantly higher in both iWAT and eWAT from wild-type female mice compared to their male counterparts at 4 months of age (S3D Fig), suggesting that *Gpc3* is a candidate escape gene from X-inactivation. It is worth noting that the expression of *Mir717*, an obesity associated microRNA [21] encoded within the third intron of the *Gpc3* gene, was unaffected in our model (S3E Fig).

### Gpc3 constrains iWAT expansion in a cell-autonomous manner during HFD feeding

To investigate the cellular mechanisms underlying *Gpc3*-mediate regulation of iWAT and eWAT expansion, we crossed *PdgfraCreER^T^* (control) and *PdgfraCreER^T^*;*Gpc3^flox^* (mutant) mice with an *mTmG* reporter strain, which labels Cre-expressing cells and their progeny with GFP while non-recombined cells express tdTomato [22], enabling lineage trace the ASPCs for quantitative analysis. Six-week-old mice received five consecutive daily tamoxifen injections, which efficiently reduced *Gpc3* mRNA in both iWAT and eWAT (S4A Fig), and then were placed on a 7-week HFD regimen (Fig 5A). In line with the non-inducible deletion model, mutant mice exhibited significantly greater weight gain than controls (Fig 5B). Strikingly, iWAT from mutant mice showed a significantly higher proportion of GFP+ traced adipocytes derived from ASPCs, along with reduced adipocyte size (Fig 5C), which could be due to the enhanced differentiation potential in *Gpc3*-deficient ASPCs. In addition, *Gpc3*-deficient (GFP+) adipocytes in the mutant iWAT were significantly smaller than the control (Tomato+) adipocytes within the same tissue microenvironment (S4B Fig). In contrast, eWAT displayed no differences in the proportion or size of traced adipocytes between genotypes (Fig 5C), consistent with a depot-specific role for *Gpc3* in iWAT under HFD. Notably, non-traced (Tomato+) adipocytes were unaffected in both depots (Fig 5C), demonstrating that *Gpc3* acts in a cell-autonomous manner to constrain tissue expansion. When using the same injection protocol

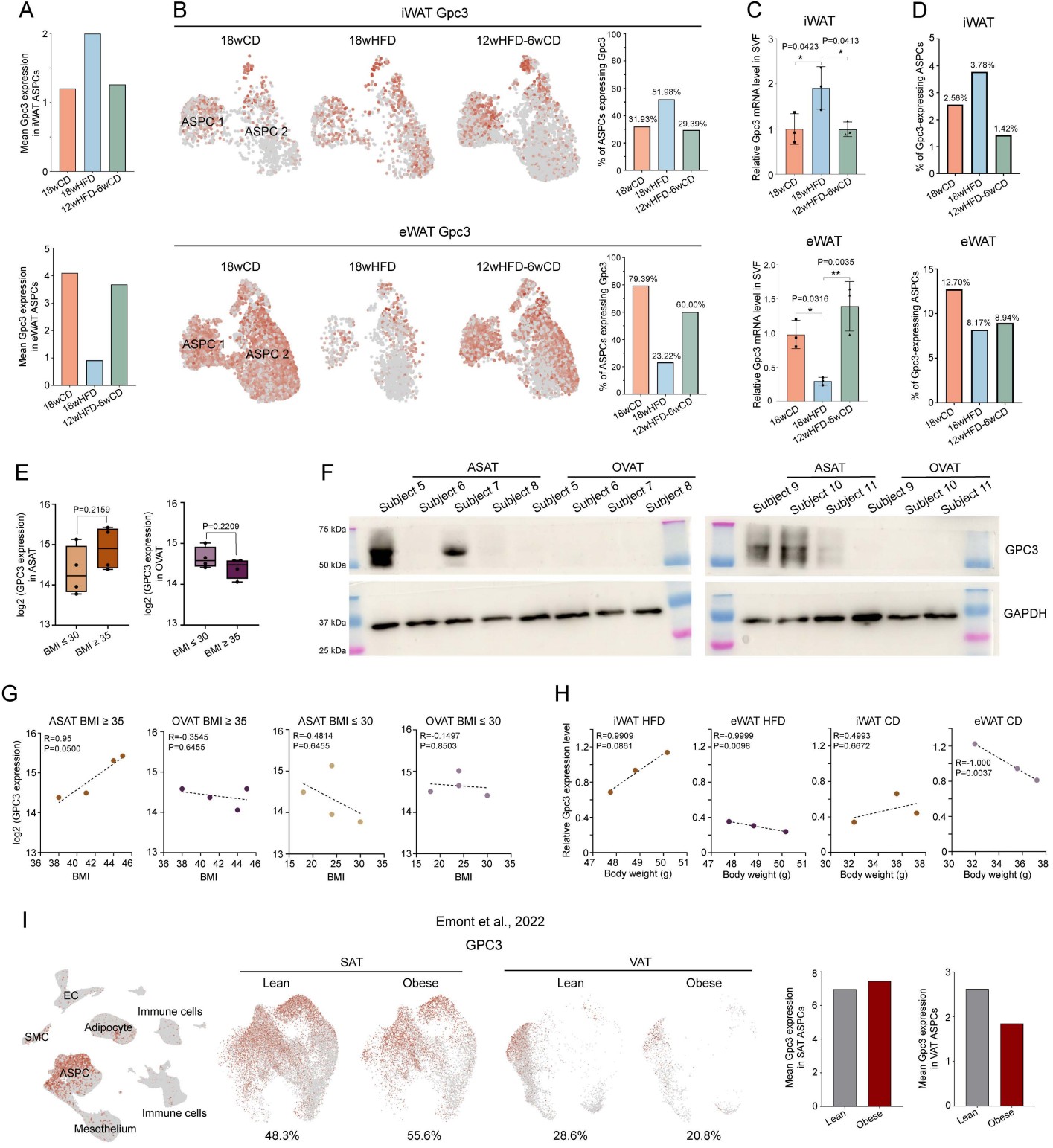

**Fig 3. Gpc3 exhibits opposing expression patterns in subcutaneous vs. visceral WAT during metabolic adaptation in mice and humans. (A)** Bar graphs show mean *Gpc3* expression in all mouse iWAT and eWAT ASPCs across dietary regimens. **(B)** UMAPs and bar charts show percentage of *Gpc3*-expressing ASPCs. **(C)** qPCR analysis of *Gpc3* mRNA level in SVFs of iWAT and eWAT from the 18wCD, 18wHFD, and 12wHFD-6wCD groups.

Mean±SD, $n=3$ animals per group, represented by a dot in the histogram. One-way ANOVA Tukey's multiple comparison test was used to determine statistical significance. Results were normalized to the 18wCD group for better visualization of the fold changes. **(D)** FACS analysis of percentage of GPC3-expressing ASPCs in SVF of iWAT and eWAT from the 18wCD, 18wHFD, and 12wHFD-6wCD groups. ASPCs were identified by PDGFRA antibody. **(E)** Box and whiskers plots show GPC3 protein levels in ASAT and OVAT of obese and non-obese human subjects. **(F)** Western blots show GPC3 protein level in paired ASAT and OVAT from obese Subject 5–11. **(G, H)** Linear regression curves of Gpc3 protein (G) and mRNA (H) expression against BMI/body weight of human donors **(G)** and mice **(H)**. Each dot represents individual human or mouse sample. *Gpc3* mRNA level in **(H)** was assessed by qPCR. **(I)** Analysis of *GPC3* expression in snRNA-seq datasets by Emont and colleagues [12]. Percentage below each UMAP of the ASPC population shows the proportion of ASPCs that expresses *GPC3*. SAT, subcutaneous adipose tissue; VAT, visceral adipose tissue. Bar charts show the mean *GPC3* expression levels in all SAT and VAT ASPCs. Numerical data of **(A)**, **(C)**, **(E)**, **(G)**, **(H)**, and **(I)** can be found in S1 Data, sheet "Fig 3". Raw images of (F) can be found in S1 Raw Images.

while maintaining the mice on CD (Fig 5D), an increasing trend in body weight gain was also observed in mutant mice, though statistical significance was not reached (Fig 5E). The only difference from the HFD condition was that mutant mice showed significantly higher proportions of GFP+ traced adipocytes in both iWAT and eWAT (Fig 5F), suggesting a diet-dependent role of *Gpc3* in regulating eWAT expansion. Of note, the actual proportions of GFP+ traced adipocytes were higher in iWAT (average 2.5% versus 1.5%) and eWAT (average 6% versus 2.5%) of HFD-fed control mice compared with CD condition, despite their visual abundance in the CD-fed mice due to smaller cell size, which is in line with a previous study showing that HFD-animals have higher rates of adipogenesis [23].

### Gpc3-deficient ASPCs in HFD-fed iWAT show enhanced differentiation potential and reduced proliferative capacity

To investigate the mechanisms underlying enhanced hyperplasia in Gpc3-deficient ASPCs, we isolated ASPCs from iWAT and eWAT of HFD-fed control and mutant mice for in vitro primary culture. By days in vitro (DIV) 7, mutant iWAT-derived adipocytes exhibited significantly greater Oil Red O-stained lipid accumulation (Fig 6A), along with increased expression of adipogenic (*Pparg*, *Cebpa*) and lipogenic (*Dgat2*) markers during differentiation (Fig 6B). These changes were accompanied by reduced BrdU incorporation (Fig 6C) and *Ccnd1* expression (Fig 6D), indicating impaired proliferative capacity. In contrast, eWAT-derived adipocytes from HFD-fed mice showed no differences in lipid accumulation (Fig 6E) or proliferative capacity (Fig 6G and 6H), despite elevated expression of adipogenic and lipogenic markers (Fig 6F). Under CD condition, control and mutant iWAT-derived adipocytes showed comparable differentiation capacity (Fig 6I) and similar *Pparg*, but not *Cebpa* or *Dgat2* expression (Fig 6J). Notably, mutant eWAT-derived adipocytes showed reduced differentiation capacity (Fig 6K) with decreased expression of adipogenic and lipogenic markers (Fig 6L). Together, these data demonstrate that Gpc3 deletion in ASPCs selectively enhances adipogenic potential while suppressing proliferation in iWAT under HFD condition.

### Gpc3 governs iWAT differentiation through canonical Wnt signaling

To elucidate the molecular mechanisms underlying the iWAT-specific effects of Gpc3, we performed comparative pathway enrichment analysis of our mouse proteomic data, focusing on pathways known to be regulated by GPC3. We found that the Wnt signaling pathway was specifically downregulated in mutant iWAT (Fig 7A), consistent with Gpc3's role as a Wnt activator in hepatocellular carcinoma [24,25]. In contrast, other pathways, including Hedgehog, Hippo, and fibroblast growth factor signaling remained unaffected or exhibited alterations only in mutant eWAT (Fig 7A). Supporting this observation, Wnt-related proteins, including total β-catenin (CTNNB1) were predominantly upregulated in mutant iWAT, while no consistent pattern was observed in eWAT (Fig 7B) (S2 Table). In adipocytes derived from iWAT ASPCs, both active and total β-catenin protein levels were significantly reduced in HFD-fed *PdgfraCre;Gpc3^flox* (mutant) mice compared to *Gpc3^flox* (control) mice (Fig 7C). In our human proteomic data, the WNT signaling pathway was upregulated in ASAT compared to OVAT under obese, but not lean condition (Fig 7D). In addition, ASAT from obese patients showed higher WNT signaling activity than that from lean patients, whereas no difference was observed in OVAT (Fig 7D).

                

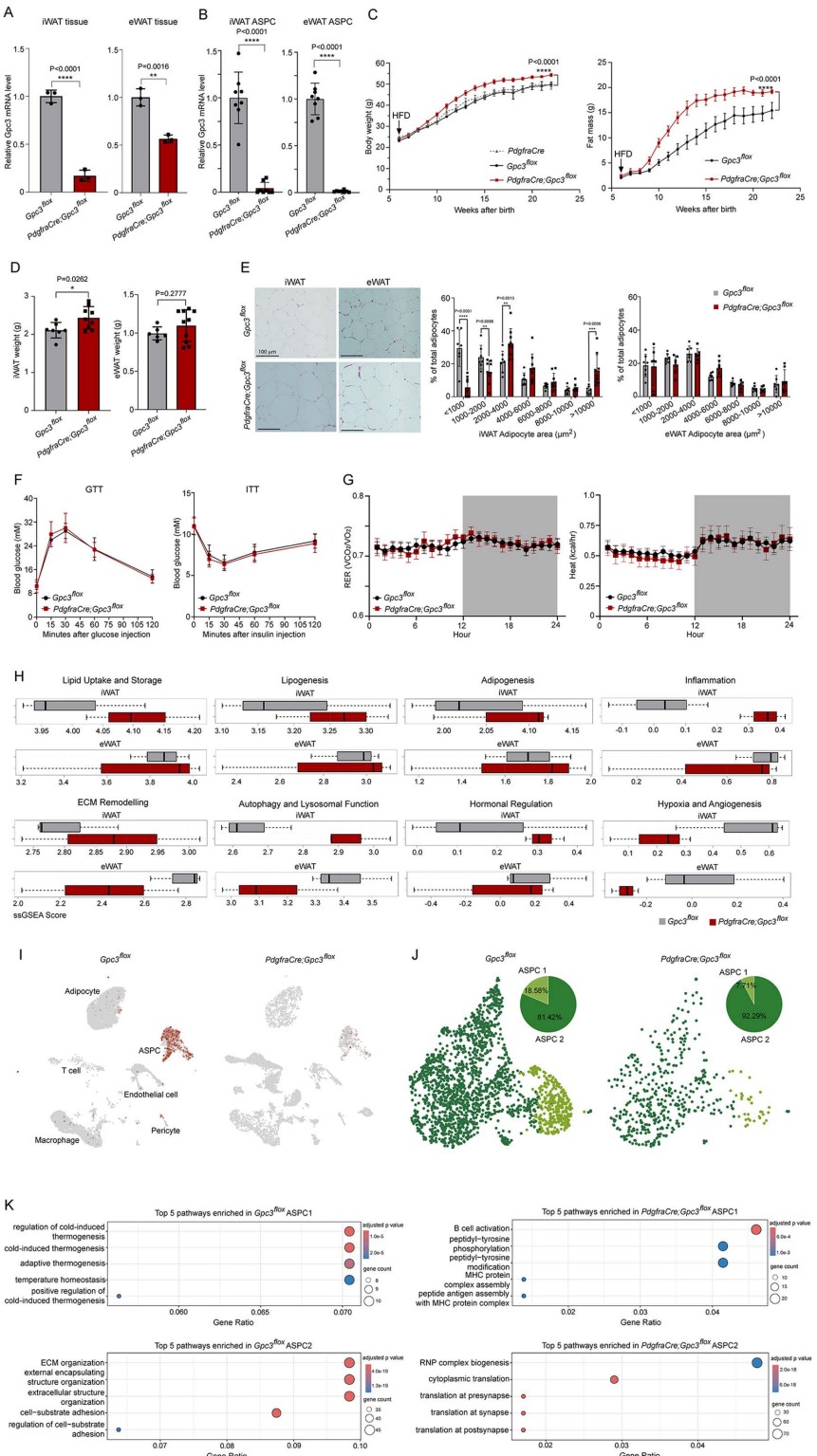

**Fig 4. Targeted deletion of Gpc3 in ASPCs selectively enhances iWAT expansion under HFD feeding condition. (A)** *Gpc3* mRNA levels in iWAT and eWAT of control and mutant mice. Mean±SD, *n*=3 for both groups, represented by a dot in the histogram. Unpaired *t* test was used to determine statistical significance. Results were normalized to the control group for better visualization of the fold changes. **(B)** *Gpc3* mRNA levels in ASPCs

isolated from iWAT and eWAT of control and mutant mice. Mean ± SD, $n = 8$ independent experiments for both groups, represented by a dot in the histogram. Unpaired $t$ test was used to determine statistical significance. Results were normalized to the control group for better visualization of the fold changes. **(C)** Body weight and fat mass (assessed by a small animal body composition analyzer) curves of control and mutant mice during HFD feeding. Mean ± SEM, $n = 8$, 13, and 19 for the *PdgfraCre*, *Gpc3^flox^* and *PdgfraCre;Gpc3^flox^* groups, respectively. Two-way ANOVA was used to determine statistical significance. $p$-values indicated in the graphs are for the independent variables. $p$-values for the interaction are 0.0663 **(B)** and 0.1211 **(C)**, respectively. **(D)** Wet weights of iWAT and eWAT of control and mutant mice by the end of HFD feeding. Mean ± SD, $n = 6$–10 animals, represented by a dot in the histogram. Unpaired $t$ test was used to determine statistical significance. **(E)** HE staining images and size-based distribution of adipocytes in iWAT and eWAT of control and mutant mice during CD feeding. Mean ± SD, $n = 7$–9 animals, represented by a dot in the histogram. Two-way ANOVA Sidak's multiple comparison test was used to determine statistical significance. All reported $p$-values compared the values of *Gpc3^flox^* and *PdgfraCre;Gpc3^flox^* groups within each adipocyte size category. **(F)** Glucose and insulin tolerance tests in HFD-fed control and mutant mice. Mean ± SD, $n = 6$ and 4 for control and mutant groups, respectively. **(G)** Respiratory exchange ratio (RER) and energy expenditure in HFD-fed control and mutant mice. Mean ± SEM, $n = 10$ and 7 for control and mutant groups, respectively. Energy expenditure was analyzed by the ANCOVA test using body weight as a covariate ($p = 0.76$). **(H)** ssGSEA scores of pathways related to adipose expansion in iWAT and eWAT. **(I)** UMAPs show *Gpc3* expression in all cell populations of iWAT from *Gpc3^flox^* and *PdgfraCre;Gpc3^flox^* groups. **(J)** UMAPs and composition of ASPC1 and 2 in *Gpc3^flox^* and *PdgfraCre;Gpc3^flox^* iWAT. **(K)** GO analysis of the top five pathways that are enriched in ASPC1 and 2 of *Gpc3^flox^* and *PdgfraCre;Gpc3^flox^* iWAT. Numerical data of **(A)**, **(B)**, **(C)**, **(D)**, **(E)**, **(F)**, and **(G)** can be found in S1 Data, sheet "Fig 4".

To determine whether the effects of Gpc3 on ASPC differentiation are Wnt-dependent, we knocked down *Axin1*, the gene that encodes the scaffold protein essential for β-catenin phosphorylation and degradation, in iWAT ASPCs from HFD-fed control and mutant mice. Successful *Axin1* knockdown (Fig 8A) increased active non-phospho β-catenin levels (Fig 8B) and reversed the enhanced adipogenic potential of mutant ASPCs (Fig 8C). Two-way ANOVA revealed a significant genotype-by-treatment interaction for adipogenesis (Fig 8C), suggesting Gpc3 influences the functional state of the destruction complex. Activation of Wnt signaling with either recombinant Wnt3a (rWnt3a) protein (Fig 8D and 8E) or CHIR99021 (a GSK3 inhibitor that stabilizes β-catenin) (Fig 8F and 8G) increased active non-phospho β-catenin levels and suppressed adipogenesis in both control and mutant ASPCs. Conversely, inhibition of Wnt signaling with XAV939 (a tankyrase inhibitor that promotes β-catenin degradation) in control and mutant ASPCs had the opposite effects (Fig 8H and 8I). The genotype-by-treatment interaction terms for adipogenesis of these three treatments were not significant (Fig 8E, 8G, and 8I), suggesting these agents act in parallel with Gpc3. Collectively, these findings establish Gpc3 as a major, though not exclusive, regulator of the canonical Wnt pathway in ASPCs under HFD feeding condition (Fig 8J). Notably, the observed reduction in mutant iWAT ASPC proliferation (Fig 6C) may reflect a trade-off with their enhanced differentiation capacity.

## Discussion

Gpc3 is best characterized as a diagnostic marker for hepatocellular carcinoma [24,26], where its elevated expression promotes tumor progression by amplifying Wnt signaling through direct interactions with Wnt and Frizzled [25,27]. In the context of adipose tissue, previous studies identified Gpc3 in fibro-inflammatory progenitors of gonadal WAT with anti-adipogenic properties [9] and as a regulator of brown adipose involution [20]. Our discovery that Gpc3 exhibits reciprocal expression dynamics between subcutaneous and visceral WAT in both mice and humans identifies this gene as an evolutionarily conserved marker of depot-specific metabolic adaptation. In murine models, Gpc3 expression was induced in iWAT but suppressed in eWAT under HFD, a pattern that was reversed upon switching back to CD. This depot-specific reciprocal regulation was recapitulated in human ASAT and OVAT samples, as well as published human snRNA-seq data from obese and non-obese individuals. These findings are in line with previous work highlighting the divergent responses of subcutaneous and visceral progenitors during adipose remodeling [10,11].

Our work identifies Gpc3 as a regulator of iWAT adipogenesis through modulation of canonical Wnt signaling. Gpc3-deficient ASPCs from iWAT exhibit enhanced adipogenic differentiation and reduced proliferative capacity, phenotypes that are recapitulated in vivo under HFD conditions and reversed by pharmacological activation of Wnt signaling. We speculate that reduced proliferation may be a direct result of enhanced differentiation. This notion aligns with previous

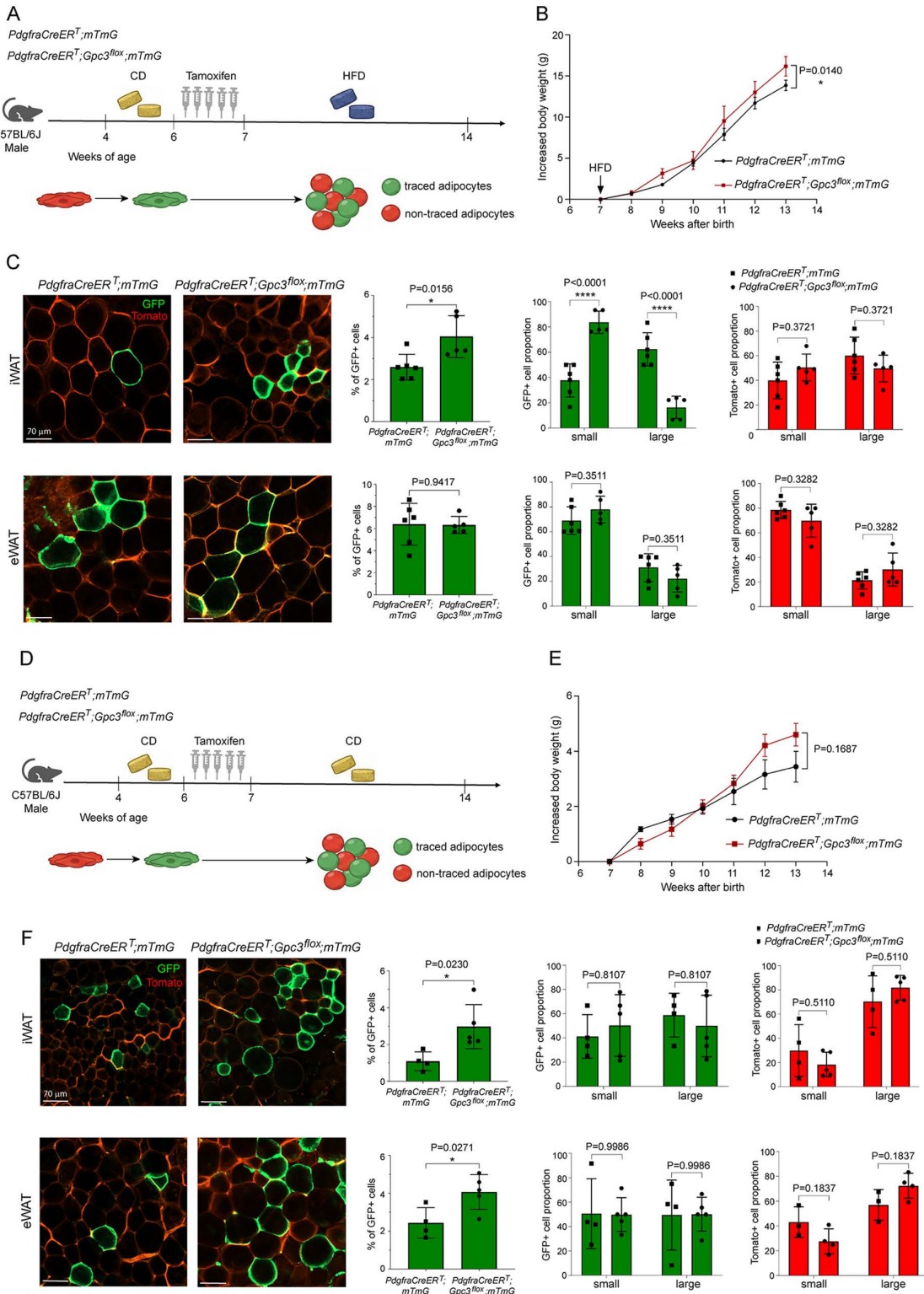

**Fig 5. Gpc3 cell-autonomously restricts iWAT expansion during HFD feeding. (A, D)** Schematic illustration of the experimental setup for HFD (**A**) and CD (**D**) feeding conditions. Created by hand. (**B**, **E**) Body weight gain of control and mutant mice during HFD (**B**) and CD (**E**) feeding. Mean ± SEM, $n = 6$, 4, 5, and 8 for the HFD and CD *PdgfraCreER^T;mTmG* and *PdgfraCreER^T;Gpc3^flox;mTmG* groups, respectively. Two-way ANOVA was used to

determine statistical significance. *p*-values for the interaction are 0.6333 **(B)** and 0.0773 **(E)**, respectively. **(C, F)** Representative confocal images and quantification of adipocyte labeling in HFD **(C)** and CD **(F)**-fed iWAT and eWAT. Mean ± SD, *n* = 3-6 animals per group, represented by a dot in the histogram. Unpaired *t* test and Two-way ANOVA Sidak's multiple comparison test were used to determine statistical significance. All reported p-values for the Two-way ANOVA Sidak's multiple comparison test compared the values of *PdgfraCreER^T;mTmG* and *PdgfraCreER^T;Gpc3^flox;mTmG* groups within each adipocyte size category. Small and large adipocytes were determined based on the median size in each comparison. HFD iWAT, small: <7,000 μm$^2$, large: ≥7,000 μm$^2$; HFD eWAT, small: <10,000 μm$^2$, large: ≥10,000 μm$^2$; CD iWAT, small: <1,500 μm$^2$, large: ≥1,500 μm$^2$; CD eWAT, small: <3,000 μm$^2$, large: ≥3,000 μm$^2$. Numerical data of **(B)**, **(C)**, **(E)**, and **(F)** can be found in S1 Data, sheet "Fig 5".

reports showing that Wnt signaling suppresses adipogenesis by inhibiting PPARγ and C/EBPα activation [28,29], as well as findings in human adipose progenitors where LRP5 knockdown enhanced adipogenic potential [30].

Our data extend the known functions of Wnt signaling in maintaining progenitor multipotency [31] by implicating Gpc3 as a modulator of Wnt pathway activity in adipose progenitors. We reveal that while the directional relationship between β-catenin levels and differentiation is preserved, the magnitude of the response is not uniform. The significant genotype-by-treatment interaction observed specifically with Axin1 siRNA, a direct intervention on the core destruction complex, indicates that the functional output of this machinery is quantitatively altered in the absence of Gpc3. This stands in contrast to the additive effects seen with upstream pathway modulators (Wnt3a, CHIR99021, XAV939). These findings support a model in which Gpc3 does not function as a simple on/off switch for Wnt signaling, but rather as a modulator that calibrates the pathway's signaling gain by fine-tuning the responsiveness of the β-catenin destruction complex.

Notably, the specificity of this mechanism to iWAT suggests that depot-intrinsic regulatory networks, including micro-environmental cues and matrix composition, may influence Gpc3-Wnt axis function. In contrast to its profound effects in iWAT, Gpc3 deletion had less impact on eWAT ASPCs. This might be explained by its reciprocal expression changing pattern: while HFD feeding upregulated Gpc3 expression in iWAT, it paradoxically downregulated it in eWAT. The depot- and diet-specific differences in adipogenic capacity observed in Gpc3-deficient ASPCs may result from the interaction between metabolic state and Gpc3 deletion. In iWAT from HFD-fed mice, loss of Gpc3 shifted progenitors toward differentiation, as evidenced by increased adipogenic and lipogenic gene expression and lipid accumulation, while simultaneously reducing proliferation. In contrast, iWAT progenitors from chow-fed mice showed minimal differences between control and mutant cells, indicating that HFD-induced metabolic stress is required for the Gpc3-induced adipogenic differences. This may arise from a combination of intrinsic depot-specific programs and environmentally induced cues. Several studies indicate that ASPCs possess depot-dependent transcriptional and proteomic profiles that govern their differentiation potential [14–16]. Exposure to HFD may further modulate these programs through altered Wnt signaling, ECM remodeling, and local inflammatory signals within the adipose niche. Such diet-induced metabolic stress can selectively influence subpopulations of progenitors, enhancing adipogenic capacity in certain depots while limiting it in others, thereby contributing to the depot-specific remodeling observed in our study.

Unexpectedly, in HFD-fed eWAT culture, adipogenic and lipogenic markers were upregulated in the mutant ASPC-derived adipocytes despite no detectable changes in differentiation or proliferative capacity were observed. In addition, in CD-fed eWAT culture, mutant progenitors exhibited reduced differentiation capacity but elevated expression of adipogenic and lipogenic markers. These uncoupling between gene expression and functional output suggests that eWAT ASPCs may exist in a transcriptionally primed state that does not readily translate into adipogenesis under obesogenic condition. Several mechanisms could account for this discrepancy, including depot-specific microenvironmental constraints such as inflammation and fibrosis, which are known to be more pronounced in eWAT and can suppress adipogenic execution even in the presence of elevated transcriptional programs [32]. Alternatively, post-transcriptional or epigenetic modifications may limit the translation of adipogenic regulators. Moreover, cell heterogeneity within the eWAT progenitor pool may result in a subset of cells driving marker expression without contributing to functional differentiation. Of note, the in vitro differentiation capacity of eWAT ASPCs is substantially lower than that of iWAT, which may further contribute to the observed

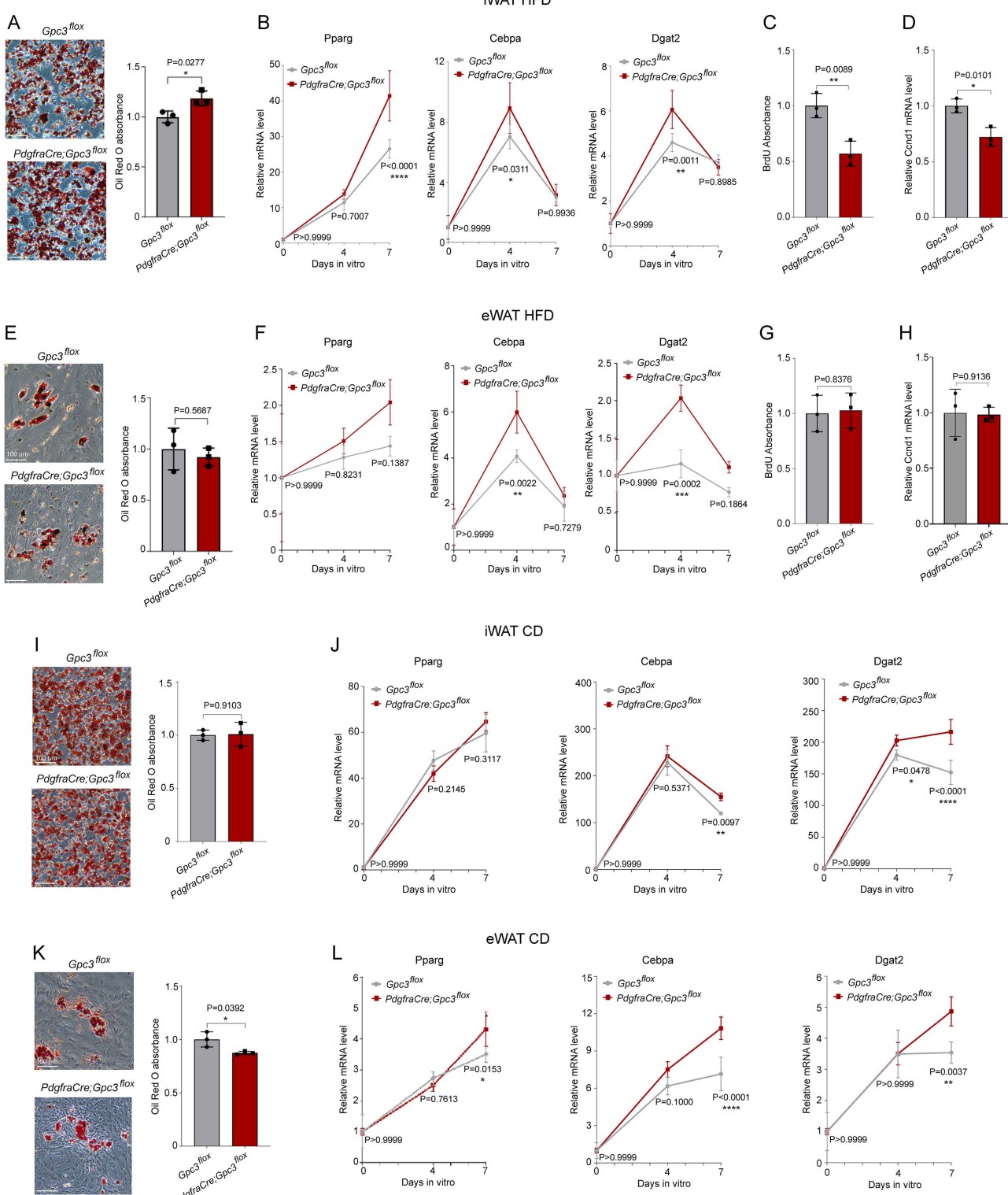

**Fig 6. Gpc3 deficiency promotes differentiation while suppressing proliferation in iWAT ASPCs under HFD condition. (A, E, I, K)** Oil Red O staining and quantification of iWAT **(A, I)** and eWAT **(E, K)** ASPC-derived adipocytes from HFD- **(A, E)** or CD-fed **(I, K)** control and mutant mice on DIV7. Mean±SD, $n = 3$ independent experiments, represented by a dot in the histogram. Unpaired $t$ test was used to determine statistical significance.

Results were normalized to the control group for better visualization of the fold changes. **(B, F, J, L)** *Pparg*, *Cebpa*, and *Dgat2* mRNA levels of control and mutant mice during HFD- **(B, F)** or CD-fed **(J, L)** iWAT **(B, J)** and eWAT **(F, L)** ASPC differentiation on DIV0, 4, and 7. Mean±SD, $n = 3$ independent experiments. Two-way ANOVA Sidak's multiple comparison test was used to determine statistical significance. All reported p-values compared the values of *Gpc3flox* and *PdgfraCre;Gpc3flox* groups at each time point. Results were normalized to the DIV0 group for better visualization of the fold changes. **(C, G)** BrdU incorporation in iWAT **(C)** and eWAT **(G)** ASPC from HFD-fed control and mutant mice on DIV2. Cells were exposed to BrdU during DIV0-2. Mean±SD, $n = 3$ independent experiments, represented by a dot in the histogram. Unpaired *t* test was used to determine statistical significance. Results were normalized to the control group for better visualization of the fold changes. **(D, H)** *Ccnd1* mRNA levels in iWAT **(D)** and eWAT **(H)** ASPC from HFD-fed control and mutant mice on DIV2. Mean±SD, $n = 3$ independent experiments, represented by a dot in the histogram. Unpaired *t* test was used to determine statistical significance. Results were normalized to the control group for better visualization of the fold changes. Numerical data of **(A)**, **(B)**, **(C)**, **(D)**, **(E)**, **(F)**, **(G)**, **(H)**, **(I)**, **(J)**, **(K)**, and **(L)** can be found in S1 Data, sheet "Fig 6".

discrepancies. Together, these findings highlight a diet- and depot-dependent complexity in Gpc3 function, in which elevated adipogenic transcriptional signatures in eWAT do not necessarily predict differentiation outcomes.

In humans, gluteofemoral subcutaneous adipose tissue accumulation ("pear-shaped" phenotype) is associated with improved glucose tolerance [33] and atheroprotection [34], whereas visceral WAT expansion is associated with increased risk of type 2 diabetes [35] and chronic heart failure [36]. Transplantation of subcutaneous adipose tissue into the visceral cavity improves insulin sensitivity in mice [6], while surgical removal or ectopic implantation of eWAT modulates age-induced insulin resistance in rats [37]. These observations highlight that intrinsic depot-specific mechanisms, rather than mere anatomical location, govern WAT functionality. Our data extend this concept by identifying Gpc3 as a molecular determinant of iWAT-specific plasticity, which agrees with the protective role of subcutaneous adipose tissue in metabolic health. Proteomic profiling of iWAT and eWAT from control and mutant mice further identified a coordinated upregulation of pathways involved in lipid metabolism, adipogenesis, inflammation, and ECM remodeling in iWAT, pointing to a multifaceted reprogramming of the adipose niche that promotes tissue expansion. These findings are consistent with studies suggesting that subcutaneous ASPCs retain greater adipogenic capacity and metabolic flexibility than their visceral counterparts [10,11].

The mechanisms driving depot-specific adipose expansion during HFD-induced obesity remain debated. Early work in mice proposed that iWAT expands primarily through hyperplasia, whereas eWAT undergoes hypertrophy after 60 days of HFD feeding, based on adipocyte size quantification and BrdU incorporation [38]. However, subsequent studies using the *AdipoChaser* mouse model revealed that both depots rely on hypertrophy during early HFD exposure (1 month), while eWAT but not iWAT switches to hyperplasia after prolonged (≥2 months) feeding [39]. These findings were corroborated by independent lineage-tracing models of *AdipoQCreER*;mTmG [40] and *PdgfraMerCreMer*;tdTomato mice [41]. In contrast, a recent study employing *PdgfraCreER^T2* knock-in mice reported that iWAT expansion after 8 weeks of HFD is driven by hyperplasia (recruitment of small adipocytes), while eWAT expands via hypertrophy [23]. Here, using a *PdgfraCreER^T* line with bacterial artificial chromosomes as a driver of the transgene coupled with *Gpc3flox* and mTmG reporter lines, we show that ASPC-specific deletion of Gpc3 selectively enhances differentiation in iWAT, but not eWAT during HFD feeding, which reconciles the "iWAT hyperplasia" model [23]. Furthermore, the different response to Gpc3 deletion under chow versus HFD feeding underscores the context-dependent nature of progenitor behavior. Intriguingly, iWAT adipocyte size was increased in non-inducible mutants (*PdgfraCre;Gpc3flox*) compared to controls after an 18-week HFD, whereas traced adipocytes from inducible mutants (*PdgfraCreER^T;Gpc3flox*) showed reduced size. This phenotypic divergence may arise from distinct compensatory mechanisms triggered by chronic versus acute Gpc3 loss. In the non-inducible model, Gpc3 deletion in ASPCs occurs early and persists throughout development and adulthood, allowing chronic adaptations that tend to favor adipocyte hypertrophy as the primary mode of tissue expansion. In contrast, the tamoxifen-inducible model enables temporal deletion of Gpc3 during six- to seven-weeks of age, which selectively captured acute progenitor recruitment and de novo adipogenesis. Moreover, the inducible model generates a mosaic tissue environment (comprising both Gpc3-null and wild-type adipocytes), which may further alter cellular behavior relative to the homogeneous mutant

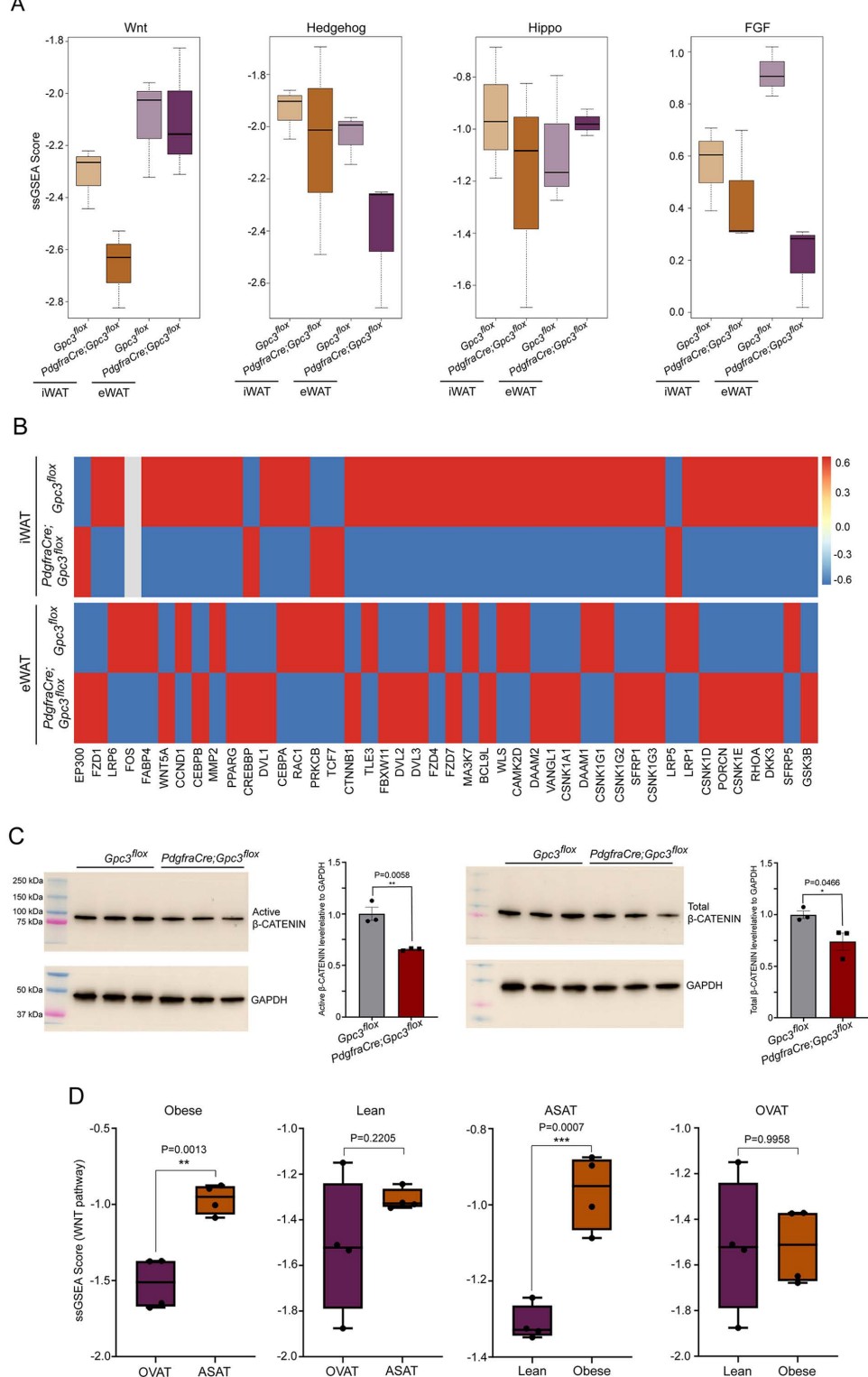

**Fig 7. iWAT-specific Wnt pathway dysregulation in Gpc3 mutant mice. (A)** ssGSEA scores of pathways regulated by GPC3 in iWAT and eWAT of HFD-fed control and mutant mice. **(B)** Heatmaps show changes in Wnt-related protein levels. Z-score was used for presentation. **(C)** Western blots show active non-phospho and total β-catenin protein levels in adipocytes derived from iWAT ASPCs of HFD-fed control and mutant mice. Mean±SD, $n=3$

independent experiments, represented by a dot in the histogram. Unpaired *t* test was used to determine statistical significance. Results were normalized to the control group for better visualization of the fold changes. **(D)** ssGSEA scores of WNT signaling pathway from human proteomic data. Numerical data of (C) can be found in S1 Data, sheet "Fig 7". Raw images of (C) can be found in S1 Raw Images.

population in non-inducible animals. These context-dependent effects suggest that the timing and duration of Gpc3 loss critically influence the balance between hyperplastic and hypertrophic remodeling in iWAT. Future studies could dissect this timing-dependent effect more precisely by using the inducible model to selectively delete Gpc3 at multiple developmental and metabolic stages, coupled with lineage tracing and functional assays, to map how Gpc3 loss influences the transition between hypertrophic and hyperplastic expansion over time.

Our integrative analysis of human adipose tissue proteomes and mouse ASPC transcriptomes reveals a conserved, depot-specific molecular identity that is both intrinsically encoded and differentially responsive to metabolic cues. In humans, we observed a strikingly non-overlapping proteomic landscape between ASAT and OVAT, consistent with previous reports of depot-specific gene expression and metabolic activity [42–44]. The pronounced enrichment of oxidative stress-related pathways in ASAT and RNA processing pathways in OVAT suggests a protective role of subcutaneous adipose tissue against systemic metabolic stress and implicates visceral adipose tissue in enhanced anabolic metabolism. In parallel, our scRNA-seq of mouse iWAT and eWAT ASPCs under various dietary conditions demonstrates that eWAT ASPCs are markedly more transcriptionally responsive to HFD, exhibiting a broader repertoire of DEGs than iWAT. Notably, the transcriptional and cellular remodeling induced by HFD was reversible upon dietary intervention, highlighting the dynamic adaptability of adipose progenitors. Together, these findings provide mechanistic insights into how depot-specific molecular programs shape adipose tissue function and remodeling across species, and suggest that targeting depot- and lineage-specific plasticity could represent a promising avenue for metabolic disease intervention.

A notable finding in our study using the *PdgfraCre* driver was the observed reduction of Gpc3 expression not only in adipose tissue but also in the brain. This indicates that the *PdgfraCre* line, while effective in targeting adipocyte precursors, may also be active in progenitor cells common to other tissues or during early embryonic development. Thus, we cannot definitively rule out that the metabolic phenotype arises from a combined effect of Gpc3 loss across multiple organs. However, several lines of evidence suggest that adipose tissue is the critical site of action for Gpc3. First, the phenotype in *PdgfraCre*;*Gpc3^flox^* mutants was characterized by increased adipogenesis in iWAT. While altered adipogenesis alone does not establish a local mechanism, our lineage-tracing and in vitro studies support a cell-autonomous contribution of the mutation within adipose progenitor cells. Second, forebrain-specific deletion of Gpc3 using Camk2a-Cre, which targets postnatal excitatory neurons of the forebrain including hypothalamic circuits implicated in energy balance [45], was insufficient to alter body weight or fat mass, arguing against a major contribution of neuronal Gpc3 to these phenotypes. Third, the *PdgfraCre*;*Gpc3^flox^* mutant mice exhibited comparable glucose tolerance, insulin sensitivity, and energy expenditure to controls. This decoupling between the adipose phenotype and the absence of broader metabolic dysfunction suggests that the obesity originates from the adipose-specific loss of Gpc3, not from its concurrent loss in other organs. Another limitation of the study is that we were unable to reliably detect GPC3 protein in adipose tissue by western blot, despite of multiple attempts with several commercially available antibodies. This limitation may reflect epitope masking due to post-translational modifications and/or insufficient antibody specificity for adipose tissue. Although this technical limitation prevented direct protein validation, the overall conclusions of our study remain unaffected, as they are supported by consistent transcriptomic and functional evidence.

In summary, our data position Gpc3 as a critical regulator of adipose tissue plasticity and distribution, providing insight into how regional fat depots differentially adapt to metabolic stress. These findings may have broader implications for understanding the mechanisms underlying healthy versus unhealthy fat expansion and for identifying targets to modulate adipose tissue function in metabolic disease. However, several lines of evidence argue that the adipose-specific deletion is the primary driver.

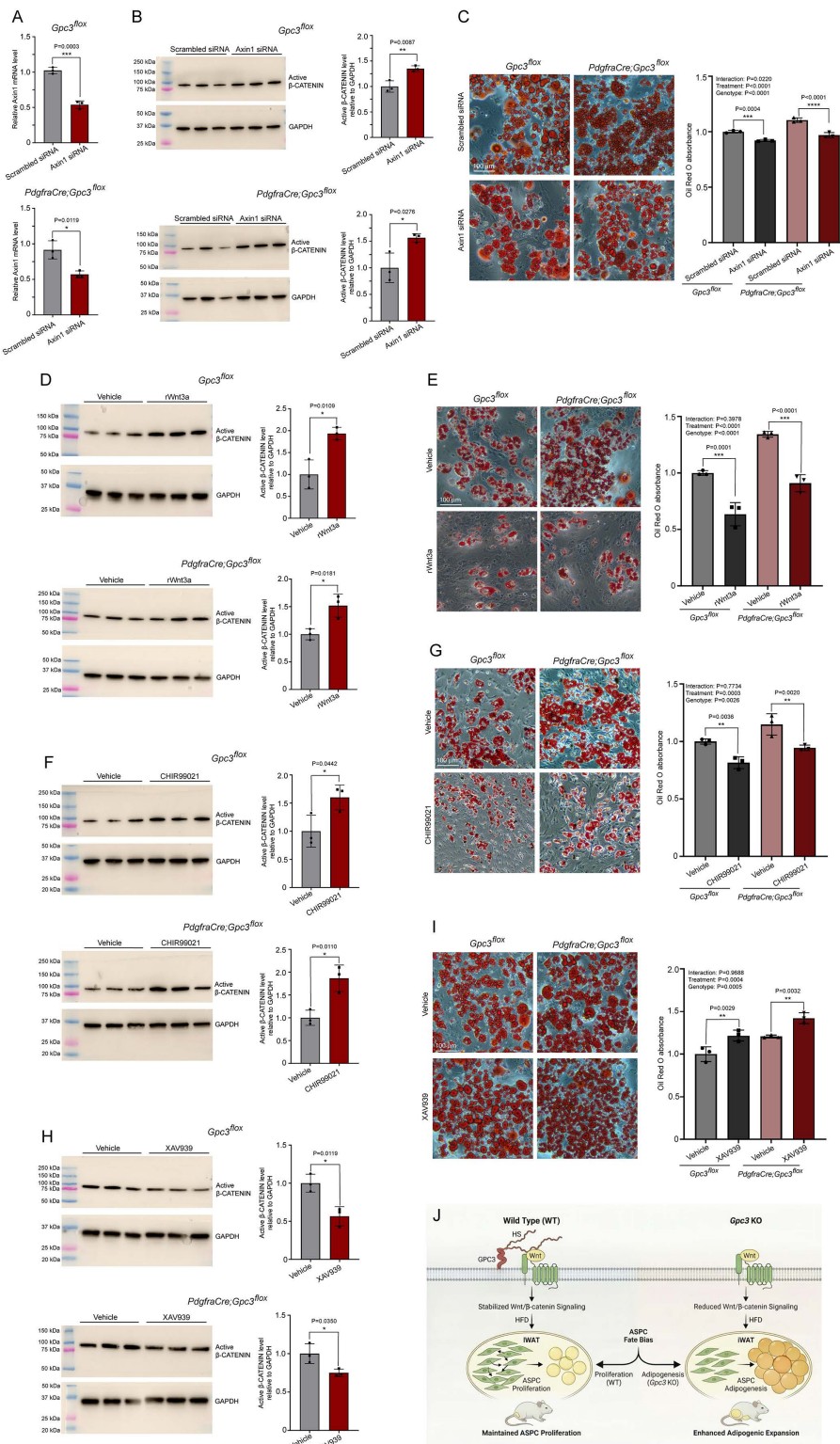

**Fig 8. Gpc3 regulates iWAT differentiation via canonical Wnt Signaling. (A)** *Axin1* mRNA levels of HFD-fed iWAT ASPCs on DIV7 from control and mutant mice. Mean ± SD, *n* = 3 independent experiments, represented by a dot in the histogram. Unpaired *t* test was used to determine statistical significance. **(B)** Western blots show active non-phospho β-catenin protein levels of scrambled and *Axin1* siRNA-treated HFD-fed iWAT ASPCs from

control and mutant mice on DIV7. siRNA was applied during DIV0-2. Mean±SD, $n = 3$ independent experiments, represented by a dot in the histogram. Unpaired $t$ test was used to determine statistical significance. Results were normalized to the scrambled siRNA group for better visualization of the fold changes. **(C)** Oil Red O staining and quantification of scrambled and *Axin1* siRNA-treated HFD-fed iWAT ASPCs from control and mutant mice on DIV7. Mean±SD, $n = 3$ independent experiments, represented by a dot in the histogram. Two-way ANOVA Sidak's multiple comparison test was used to determine statistical significance. Results were normalized to the vehicle-treated control group for better visualization of the fold changes. **(D)** Western blots show active non-phospho β-catenin protein levels in vehicle (PBS with 1% BSA)- and rWnt3a-treated iWAT ASPCs from HFD-fed mutant mice on DIV7. rWnt3a treatment was applied during DIV0-3. Mean±SD, $n = 3$ independent experiments, represented by a dot in the histogram. Unpaired $t$ test was used to determine statistical significance. Results were normalized to the vehicle group for better visualization of the fold changes. **(E)** Oil Red O staining and quantification of vehicle- and rWnt3a-treated iWAT ASPCs from HFD-fed control and mutant mice on DIV7. Mean±SD, $n = 3$ independent experiments, represented by a dot in the histogram. Two-way ANOVA Sidak's multiple comparison test was used to determine statistical significance. Results were normalized to the vehicle-treated control group for better visualization of the fold changes. **(F)** Western blots show active non-phospho β-catenin protein levels in vehicle (0.004% DMSO)- and CHIR99021-treated iWAT ASPCs from HFD-fed control and mutant mice on DIV7. CHIR99021 treatment was applied during DIV0-7. Mean±SD, $n = 3$ independent experiments, represented by a dot in the histogram. Unpaired $t$ test was used to determine statistical significance. Results were normalized to the vehicle group for better visualization of the fold changes. **(G)** Oil Red O staining and quantification of vehicle- and CHIR99021-treated iWAT ASPCs from HFD-fed control and mutant mice on DIV7. Mean±SD, $n = 3$ independent experiments, represented by a dot in the histogram. Two-way ANOVA Sidak's multiple comparison test was used to determine statistical significance. Results were normalized to the vehicle-treated control group for better visualization of the fold changes. **(H)** Western blots show β-catenin protein levels in vehicle (DMSO)- and XAV939 (0.4 μM)-treated iWAT ASPCs from HFD-fed control and mutant mice on DIV7. XAV939 treatment was applied during DIV0-3. Mean±SD, $n = 3$ independent experiments, represented by a dot in the histogram. Unpaired $t$ test was used to determine statistical significance. Results were normalized to the vehicle group for better visualization of the fold changes. **(I)** Oil Red O staining and quantification of vehicle- and XAV939-treated iWAT ASPCs from HFD-fed control and mutant mice on DIV7. Mean±SD, $n = 3$ independent experiments, represented by a dot in the histogram. Unpaired $t$ test was used to determine statistical significance. Results were normalized to the vehicle-treated control group for better visualization of the fold changes. **(J)** Schematic of the depot-specific regulation of adipogenesis by Gpc3 via the canonical Wnt pathway. Created by hand. Numerical data of **(A)**, **(B)**, **(C)**, **(D)**, **(E)**, **(F)**, **(G)**, **(H)**, and **(I)** can be found in S1 Data, sheet "Fig 8". Raw images of **(B)**, **(D)**, **(F)**, and **(H)** can be found in S1 Raw Images.

## Materials and methods

### Human adipose tissue samples

Paired ASAT and OVAT collection was approved by the Peking University Third Hospital Medical Science Research Ethics Committee (M2024238) and performed in compliance with all relevant ethical regulations, with written informed consent obtained from all participants. The study was conducted in accordance with the principles expressed in the Declaration of Helsinki. General information of the participants was listed in S1 Table. Excess adipose tissue was collected during standard surgical procedures, with no additional interventions required.

### Mouse adipose tissue samples

All animal procedures were conducted in accordance with protocols approved by the Institutional Animal Care and Use Committees (IACUC) of Peking University (Protocol #Psych-XieM-2) and the Chinese Institute for Brain Research (Protocol #CIBR-IACUC-035), ensuring full compliance with ethical guidelines for animal research. The study was conducted in accordance with the Guidelines for the ethical review of laboratory animal welfare People's Republic of China National Standard GB/T 35892-2018. *Gpc3^flox^* mice (T052331) were obtained from GemPharmatech (Nanjing, China). C57BL/6J (000664), *PdgfraCre* (013148), *PdgfraCreER^T^* (018280), and mTmG (007676) mice were obtained from the Jackson Laboratory. Six-week-old mice were fed with CD (1010097, Jiangsu Xietong Pharmaceutical Bio-engineering Co., Jiangsu, China) or 60% HFD (D12492, Research Diet). All mice were maintained in a temperature-controlled environment with regulated humidity, standard 12:12 hour light/dark cycle, and free access to food and purified water. For genetic lineage tracing, tamoxifen (Sigma-Aldrich, T5648) was prepared in corn oil (Solarbio, C7030) and administered via intraperitoneal injection at 2 mg/day for 5 consecutive days. Fat mass was measured using a small animal body composition analyzer (QMR12-060H-I, Niumag Analytical Instrument Corporation, Suzhou, China). Male mice ranging from 6 to 24 weeks of age were used in the present study.

## Human and mouse adipose sample preparation for DIA quantitative proteomics and data analysis

Human and mouse adipose tissues were snap-frozen in liquid nitrogen post-excision and stored at −80 °C. Tissues were pulverized in liquid nitrogen and lysed in SDT buffer (4% SDS, 100 mM Tris-HCl, pH 7.6, 100 mM NaCl, 100 mM DTT). Lysates were sonicated on ice for 5 min, heated at 95 °C for 10 min, and centrifuged at 12,000$g$ for 15 min at 4 °C. Supernatants were alkylated with 100 mM iodoacetamide (Sigma-Aldrich) in the dark at room temperature (RT) for 1 hour. Proteins were precipitated with four volumes of pre-chilled acetone at −20 °C for 2 h, pelleted at 12,000$g$ for 15 min at 4 °C, washed with ice-cold acetone, and dissolved in 8 M urea with 100 mM triethylammonium bicarbonate (pH 8.5; Sigma-Aldrich). Protein concentrations were measured using the Bradford assay (Bio-Rad) with bovine serum albumin as standard. Proteins (20 µg) were digested with trypsin (Promega) at a 1:50 ratio in 100 mM triethylammonium bicarbonate at 37 °C for 4 h, then overnight. Peptides were acidified to pH < 3 with formic acid, desalted on C18 Sep-Pak columns (Waters), eluted in 70% acetonitrile with 0.1% formic acid, and lyophilized. Peptides were reconstituted in 0.1% formic acid and analyzed by DIA on a Thermo Fisher Orbitrap Astral mass spectrometer with a Vanquish Neo UHPLC system. Separation used a PepMap Neo C18 column (150 µm × 15 cm, 2 µm; Thermo Fisher Scientific) with a C18 pre-column (5 mm × 300, 5 µm) at 50 °C, using a 22-min gradient (4%–99% mobile phase B: 80% acetonitrile, 0.1% formic acid). The mass spectrometer operated in DIA mode (spray voltage 1.9 kV, ion transfer tube 290 °C), scanning precursor ions ($m/z$ 380–980) at 240,000 resolution and fragment ions ($m/z$ 150–2000) at 80,000 resolution, with 300 DIA windows, 25% normalized collision energy, and 3 ms maximum injection time.

Raw DIA data were processed using DIA-NN (v1.8) against a UniProt database, with 10 ppm precursor and 0.02 Da fragment mass tolerances, cysteine alkylation, and up to one missed cleavage. Peptide Spectrum Matches were filtered at 99% confidence, with FDR < 0.01). Protein quantification data from adipose tissue samples were imported from a.csv file and processed to remove duplicate gene entries, retaining unique gene identifiers. All intensity values were transformed using $\log_2(x+1)$ to stabilize variance and normalize the data distribution. For paired human sample analysis, protein abundance differences between ASAT and OVAT were evaluated using data from four lean and four obese individuals. Protein enrichment in ASAT or OVAT was determined based on the direction and magnitude of expression differences between paired samples. For comparisons between lean and obese individuals in humans, and between CD and HFD groups in mice, statistical significance was assessed using unpaired two-sample $t$-tests. Analyses were restricted to proteins with sufficient non-missing values and non-zero variance. Differentially abundant proteins were annotated based on fold change and $p$-value thresholds, and categorized as enriched in the respective groups. Differentially enriched proteins were subjected to Gene Ontology (GO) enrichment analysis. Gene symbols were mapped to Entrez Gene IDs using a standard annotation database. Enrichment analysis for biological process terms was performed using inclusive $p$-value and $q$-value thresholds to ensure comprehensive coverage of significant terms. The top enriched GO terms were visualized in a dot plot to highlight key biological processes. Single-sample Gene Set Enrichment Analysis (ssGSEA) was performed to quantify pathway activity scores across individual samples. Enrichment scores were visualized as boxplots stratified by group.

The human proteomic study was designed as a targeted, hypothesis-generating investigation to identify the most robust and large-magnitude proteomic differences, rather than a fully powered, population-level discovery effort. We acknowledge that our sample size ($n = 4$ per group) presents a challenge for statistical power; however, this design is strategically focused on uncovering only the most pronounced biological signals, which are more likely to be biologically relevant and less likely to be false positives, even within a smaller cohort.

## Mouse SVF cell isolation

iWAT and eWAT were dissected, washed twice with HBSS, then minced and digested in 1 mg/ml collagenase type I (Sigma-Aldrich, C0130) dissolved in digestion buffer (0.1 M HEPES-Na, 120 mM NaCl, 50 mM KCl, 5 mM D-glucose, 1 mM CaCl$_2$, 1.5% BSA) for 45 min at 37°C with continuous shaking (150 rpm). Digestion was quenched by adding an

equal volume of growth medium (Advanced DMEM/F-12 [Gibco, 12634028] supplemented with 10% FBS, 1% Gluta-MAX [Thermo, 35050061], 1% MEM NEAA [Thermo, 11140050], and 1% penicillin-streptomycin [Gibco, 15140122]). The suspension was sequentially filtered through 100-μm and 40-μm cell strainers (Fisher Scientific), with centrifugation (450$g$ for 10 min) between steps. Erythrocyte lysis was performed by resuspending the pellet in red blood cell lysis buffer (5 min, 37°C), followed by neutralization with growth medium and final filtration with the 40-μm strainer.

## Nuclei isolation for snRNA-seq

Following dissection, iWAT was minced with scissors in 500 μl of ice-cold Nuclei Isolation Buffer, which consists of 250 mM sucrose, 10 mM HEPES, 1.5 mM $MgCl_2$, 10 mM KCl, 0.001% NP-40, 0.2 mM DTT, and 1 U/μl RNase inhibitor in DEPC-treated water. The minced tissue was then homogenized using a 2 ml Dounce homogenizer and filtered through a 70-μm cell strainer. The homogenate was centrifuged at 500$g$ for 5 min at 4 °C to pellet the nuclei. The resulting nuclei pellet was resuspended in a Nuclei Resuspension Buffer containing 2% BSA, 1.5 mM $MgCl_2$, and 1 U/μl RNase inhibitor in PBS. All buffers were sterile-filtered before use.

## scRNA-seq and snRNA-seq library preparation and sequencing

Isolated nuclei were stained with 10 μg/ml 7-AAD (Invitrogen, A1310) and sorted using a BD FACSMelody 4-Way Cell Sorter (BD Biosciences) equipped with a 100-μm nozzle. Isolated cells were stained with 7-AAD (5 μg/ml) for viability assessment prior to FACS sorting. Cell and nuclei concentration was determined using a Countstar Rigel S5 Automated Cell Counter. For single-cell/nucleus capture, approximately 20,000 viable cells/nuclei were loaded onto a 10× Genomics Chromium Controller using the Single Cell 3′ Reagent Kit v3.1 (10× Genomics) with 3′ v3 chemistry. GEM-RT was conducted at 53 °C for 45 min followed by 85 °C for 5 min, with a 4°C hold. cDNA was recovered through emulsion lysis and purified using DynaBeads with SPRIselect reagent (Thermo Scientific), followed by PCR amplification. Library preparation involved enzymatic fragmentation, size selection, and sequential addition of P5/P7 adapters and i5/i7 indexes via end repair, A-tailing, and ligation. Final libraries were quantified and pooled for 150 bp paired-end sequencing on an Illumina NovaSeq 6000, targeting ~50,000 raw reads per cell/nucleus. cDNA quality was verified at each stage using an Agilent 2100 Bioanalyzer.

## scRNA-seq and snRNA-seq data analysis

Raw FASTQ files were aligned to the mm10 reference genome (Ensembl annotation) using Cell Ranger 7.0.0 with intronic reads included. Initial gene count matrices were processed through CellBender 0.2.2 to remove ambient RNA contamination and empty droplets [46]. Subsequent quality control was performed in Seurat 4.3.0 [47], applying the following filtration criteria: 200−7,500 detected genes, 500−75,000 UMIs, and <10% mitochondrial RNA content. DoubletFinder 2.0.3 was used to exclude cell doublets [48]. To address sequencing depth heterogeneity, we performed comprehensive data preprocessing. Initial normalization was conducted using the LogNormalize method, scaling each cell's gene counts by its total UMI count (factor = 10,000) followed by natural log transformation. Highly variable genes were identified through variance stabilizing transformation with mean-variance modeling. Expression values were subsequently scaled and centered using the ScaleData function (mean = 0, standard deviation = 1). We also applied sctransform to regularize the data using a negative binomial model, reducing technical artifacts while preserving biological variation [49,50]. To integrate multiple datasets and address batch effects, we applied the Harmony algorithm to analyze and adjust principal components, aligning data across different batches. This approach effectively reduced technical variations caused by experimental conditions and processing times while maintaining biological heterogeneity. The method enabled robust integration of cells from diverse datasets while preserving biologically relevant differences [6,51]. DEGs were identified using Seurat's "FindMarkers" function with the following thresholds: |log2FC| > 0.25, adjusted $p$-value < 0.05, and expression in >25% of cells. Cell type annotation was performed by integrating biological knowledge, established cell-type-specific markers, and

top marker genes from the PanglaoDB database. The same "FindMarkers" function was applied to detect DEGs across different experimental conditions. Gene set enrichment analysis was subsequently performed using the 'gsva' function from the GSVA package to compute enrichment scores for each sample.

## RNA isolation and qPCR analysis

RNA was extracted with the TRIzol reagent (Invitrogen, 15596018). For cDNA synthesis, 1 µg of total RNA was reverse transcribed using the H Minus First Strand cDNA Synthesis Kit (Thermo Scientific, K1652). Quantitative PCR was performed using PowerUp SYBR Green Master Mix (Applied Biosystems, A25742) on a CFX96 Real-Time PCR Detection System (Bio-Rad). Gene expression levels were normalized to Ywhaz reference gene using the ΔΔCt method. For Mir717 quantification, total RNA was reverse-transcribed with a kit using a miRNA-specific stem-loop primer and a U6 snRNA primer (Servicebio). The resulting cDNA was subjected to qPCR with SYBR Green Master Mix and Mir717-specific primers. All samples were normalized to U6 snRNA.

The following mouse-specific primer sequences were used:

Gpc3: 5′-GGAGCAAGACGTGACCTGAA-3′, 5′-GCATACGGCCACAGTCCTTA-3′

Pparg: 5′-GTACTGTCGGTTTCAGAAGTGCC-3′, 5′-ATCTCCGCCAACAGCTTCTCCT-3′

Cebpa: 5′-CAAGAACAGCAACGAGTACCG-3′, 5′-GTCACTGGTCAACTCCAGCAC-3′

Dgat2: 5′- CTGTGCTCTACTTCACCTGGCT-3′, 5′- CTGGATGGGAAAGTAGTCTCGG-3′

Ccnd1: 5′-AACTACCTGGACCGCTTCCT-3′, 5′- CCACTTGAGCTTGTTCACCA-3′

Ywhaz: 5′-CAGTAGATGGAGAAAGATTTGC-3′, 5′-GGGACAATTAGGGAAGTAAGT-3′

Mir717 forward: 5′-ACACCTCAGCTGGGCTCAGACAGAGATACC-3′

Mir717 stem-loop RT primer: 5′-CTCAACTGGTGTCGTGGAGTCGGCAATTCAGTTGAGAGAGAGG-3′

Universal reverse primer: 5′-TGGTGTCGTGGAGTCG-3′

U6 forward primer:5′-CTCGCTTCGGCAGCACA-3′

U6 reverse primer: 5′-AACGCTTCACGAATTTGCGT-3′

## FACS

SVF cells isolated from iWAT and eWAT of C57BL/6J mice from the three dietary groups were resuspended in 2% FBS/PBS containing anti-mouse CD16/CD32 Fc Block (clone 2.4G2, BD) and incubated on ice for 20 min. Cells were then incubated with CD140a (PDGFRA) Antibody-APC (Thermo Fisher, 17-1401-81) and GPC3 (Thermo, MA5-17083) antibody while rotating at 4 °C for 30 min. Following incubation, cells were washed three times with 2% FBS/PBS and stained with 7-AAD to exclude dead cells. Samples were then subjected to cell analysis and sorting using a Beckman Coulter MoFlo XDP flow cytometer. The samples were loaded and run at a set flow rate, while the instrument's threshold and scatter parameters were adjusted to identify individual cells. Forward scatter (FSC) and side scatter were used to assess cell size and internal complexity. Doublets were identified and excluded by comparing the height and width parameters of FSC.

## Western blot

Protein extraction was performed by adding 0.3 mL of 100% ethanol to the interphase and lower organic phase obtained from TRIzol lysates, followed by centrifugation at 2,000g for 5 min at 4 °C. The upper phase was collected and proteins were precipitated with isopropanol (12,000g, 10 min, 4 °C). The resulting protein pellet was washed sequentially with: (1)

0.3 M guanidine hydrochloride (Solarbio, G8070) in 95% ethanol (two washes), (2) 100% ethanol (one wash), and then air-dried. Proteins were denatured in 200 µL of 1% SDS (Solarbio, S8010) at 50°C. Protein concentration was determined using a BCA assay kit (Solarbio, PC0020). For immunoblotting, 15 µg of protein was resolved by SDS-PAGE and transferred to PVDF membranes (Bio-Rad, 1620177). Membranes were developed with chemiluminescent HRP substrate (Millipore, WBKLS05000) and visualized using an Amersham ImageQuant 800 system. Band intensity quantification was performed using ImageJ software (v1.53k). The following antibodies were used: GPC3 (Santa Cruz, sc-390587), non-phospho β-Catenin (CST, #8814), total β-Catenin (CST, #9562), GAPDH (CST, #2118).

## SVF cell primary culture for adipogenic differentiation

Isolated cells were resuspended in basal medium and plated in culture dishes. Prior to plating, viable cell numbers were quantified using a hemocytometer (Neubauer chamber) with trypan blue exclusion. Cells were then seeded in 12-well plates at $7.5 \times 10^4$ cells/mL in a final volume of 2 mL per well. Upon reaching ~90% confluency, differentiation was initiated by replacing the medium with differentiation cocktail containing: advanced DMEM/F12, 5% FBS, 1% GlutaMAX, 1% NEAA, 1% P/S, 1% ITS (Gibco, 41400045), 33 µM d-biotin (Sigma-Aldrich, B4639), 17 µM pantothenate (Sigma-Aldrich, P5155), 0.5 mM IBMX (Sigma-Aldrich, I5879), 1 µM dexamethasone (Sigma-Aldrich, D4902), 1 µM rosiglitazone (Sigma-Aldrich, R2408), and 2 nM T3 (Sigma-Aldrich, T6397). After 4 days, cells were switched to maintenance medium (advanced DMEM/F12 with 2% FBS, 33 µM d-biotin, 17 µM pantothenate, 1 µM dexamethasone, 10 µg/mL insulin, 2 mM GlutaMAX, 0.1 mM NEAA, and 100 U/mL P/S) for 3 days. Cells were collected on DIV 0, 4, and 7 for analysis. GSK3 inhibitor CHIR99021 (MedChemExpress, HY-10182) and Tankyrase inhibitor XAV939 (MedChemExpress, HY-15147) were applied from DIV 0–7.

## Oil Red O staining and quantification

Cells were washed three times with PBS to remove culture medium, followed by fixation with 4% PFA for 10 min at RT. After fixation, cells were stained with Oil Red O solution (Sigma-Aldrich, O1391) for 1 h at RT with gentle agitation. Unbound stain was removed by thorough washing with distilled water until the effluent became clear. Stained lipid droplets were imaged using an Axio Observer inverted microscope (Zeiss). Bound Oil Red O was then extracted with 100% isopropanol (15 min, RT) and absorbance measured at 500 nm using a microplate reader.

## Hematoxylin and eosin (HE) staining of adipose samples and adipocyte quantification

Fresh iWAT and eWAT samples were fixed in 4% paraformaldehyde (PFA) overnight at 4 °C, followed by PBS washing. After sequential ethanol dehydration, tissues were paraffin-embedded and sectioned at 5 µm thickness. Sections were stained with HE solution (Servicebio, G1003) and imaged using an Axio Observer inverted microscope (Zeiss). Adipocyte size quantification was performed using AdipoCount software (http://www.csbio.sjtu.edu.cn/bioinf/AdipoCount/) through the following computational steps: (1) image thresholding to generate binary images and segment adipocyte membranes based on pixel intensity; (2) edge detection with histogram equalization to enhance contrast; and (3) post-processing combining thresholding and edge detection outputs for connected region analysis to determine individual adipocyte areas.

## Immunofluorescence staining of adipose samples

Paraffin-embedded tissue sections were deparaffinized, rehydrated, and subjected to heat-induced antigen retrieval in a microwave oven. After blocking endogenous peroxidase and non-specific binding sites, sections were incubated overnight with primary antibody at 4 °C. Detection was achieved using an HRP-conjugated secondary antibody followed by tyramide signal amplification. Nuclei were counterstained with DAPI, autofluorescence was quenched, and slides were mounted with anti-fade mounting medium. Following antibodies were used: DPP4 (Abcam, #187048), PDGFRB (CST, #3169).

## Whole-body metabolic status assessment

Indirect calorimetry was conducted using a comprehensive laboratory animal monitoring system (Columbus Instruments, Columbus, OH). After a 3-day acclimation period in the metabolic cages, data were collected from individually-housed mice over a 24-hour automated recording session. Mice had *ad libitum* access to food and water throughout the acclimation and data collection phases. Measurements were collected continuously and are presented as hourly averages across the light and dark phases. The experiments were conducted in adult mice fed a HFD starting at 6 weeks of age for 18 weeks (final age ~4 weeks).

## Glucose tolerance test

After a 6-hour fast with free access to water, mice received an intraperitoneal injection of 50% (w/v) glucose solution at a dose of 2 g/kg body weight. Blood glucose levels were measured from tail vein samples at 0, 15, 30, 60, 90, and 120 min post-injection using a handheld glucometer (Accu-Chek, Roche Diagnostics, IN, USA).

## Insulin tolerance test

Following a 6-hour fast, mice received an intraperitoneal injection of human insulin (Eli Lilly, Cat# 00002821501) at 1 U/kg body weight. Blood glucose was measured from tail vein samples at 0 (baseline), 15, 30, 60, 90, and 120 min post-injection using a handheld glucometer (Accu-Chek, Roche Diagnostics, IN, USA).

## Mouse adipose tissue whole-mount imaging

Mice were anesthetized and perfused with PBS prior to euthanasia. iWAT and eWAT were dissected and fixed overnight in 1% PFA at 4 °C. Following fixation, tissues were washed in PBST (PBS containing 0.01% Tween-20), embedded in 5% low-melting agarose, and sectioned into 100–400 µm slices using a Leica VT1200 Vibratome. Samples were imaged with an Airyscan 2 LSM 900 confocal microscope (ZEISS) and images were processed using the Imaris software (Bitplane version 9.0.1).

## BrdU assay

iWAT and eWAT SVF cells were stimulated with induction medium for 12 and 24 hours, respectively, before labeling with 10 µM BrdU (Roche, 11647229001) for 30 min. Cells were then fixed for DNA denaturation using the FixDenat solution, followed by incubation with anti-BrdU-POD antibody for 90 min at RT. After three washes with PBS-based buffer, tetramethylbenzidine substrate was added, and the reaction was incubated at RT until sufficient color development. Absorbance at 450 nm was measured using a microplate reader, with signal intensity reflecting DNA synthesis as a proxy for proliferation.

## Statistics

Statistical analyses were performed using GraphPad Prism version 8.0.0. Statistical methods were specified in Figure legends. The level of statistical significance was assigned as $*p < 0.05$, $**p < 0.01$, $***p < 0.001$, $****p < 0.0001$.

## Supporting information

**S1 Fig. scRNA-seq analysis of the SVF from iWAT and eWAT. (A)** Quality metrices showing unique molecular identifier (UMI) count, number of detected genes, and percentage of mitochondrial sequencing reads. **(B)** UMAP of all cell types in the SVF of iWAT and eWAT across five dietary groups. **(C)** Heatmap depicting marker gene expression levels across identified cell types. **(D)** Expression patterns of *Dpp4*, *P16*, *Pdgfrb,* and *Icam1* in ASPC subpopulations. (S1_Fig.TIF)

**S2 Fig. DPP4 (A) and PDGFRB (B) staining in eWAT of the three dietary conditions.** Mean±SD, $n=4$ animals, represented by a dot in the histogram. One-way ANOVA Tukey's multiple comparison tests was used to determine statistical significance. Numerical data of **(A)** and **(B)** can be found in S1 Data, sheet "S2 Fig".
(S2_Fig.TIF)

**S3 Fig. Tissue-specific Gpc3 expression in the constitutive knockout model. (A)** Food intake and ambulation in HFD-fed control and mutant mice. Mean±SEM, $n=10$ and 7 for control and mutant groups, respectively. **(B)** *Gpc3* mRNA levels in pancreas and brain from HFD-fed control and mutant mice. Mean±SD, $n=3$ for both groups. Unpaired *t* test was used to determine statistical significance. **(C)** Body weight and fat mass curves of *Gpc3$^{flox}$* and *Camk2aCre;Gpc3$^{flox}$* mice during HFD feeding. Mean±SEM, $n=9$ and 8 for the *Gpc3$^{flox}$* and *Camk2aCre;Gpc3$^{flox}$* groups respectively. **(D)** *Gpc3* mRNA levels in iWAT and eWAT from CD-fed 4-month-old wild-type mice. Mean±SD, $n=3$ for both groups. Unpaired *t* test was used to determine statistical significance. **(E)** Mir717 level in iWAT and eWAT from HFD-fed control and mutant mice. Mean±SD, $n=3$ for both groups. Unpaired *t* test was used to determine statistical significance. Numerical data of **(A)**, **(B)**, **(C)**, **(D)**, and **(E)** can be found in S1 Data, sheet "S3 Fig".
(S3_Fig.TIF)

**S4 Fig. Gpc3 expression in the inducible knockout model. (A)** *Gpc3* mRNA levels in iWAT and eWAT from *PdgfraCreER$^T$;mTmG* and *PdgfraCreER$^T$;Gpc3$^{flox}$;mTmG* mice one day after tamoxifen injection. Mean±SD, $n=3$ for both groups. Unpaired *t* test was used to determine statistical significance. **(B)** Comparison of GFP+ and Tomato+ adipocyte size in iWAT of HFD-fed *PdgfraCreER$^T$;Gpc3$^{flox}$;mTmG* mice. Each dot represents individual adipocytes pooled from 5 independent animals. Mann–Whitney test was used to determine statistical significance. Numerical data of **(A)** and **(B)** can be found in S1 Data, sheet "S4 Fig".
(S4_Fig.TIF)

**S1 Table. Demographic information of donors contributing adipose tissue specimens.**
(S1_Table.XLSX)

**S2 Table. Numerical values for proteins listed in Fig 7B.**
(S2_Table.XLSX)

**S1 Data. Excel spreadsheet showing all the raw numerical values displayed in relevant figure panels.**
(S1_Data.XLSX)

**S1 Raw Images. Raw western blot images displayed in relevant figure panels.**
(S1_Raw_Images.PDF)

## Acknowledgments

The authors thank Shuo Zhang for animal husbandry.

## Author contributions

**Conceptualization:** Carlos F. Ibáñez, Meng Xie.

**Formal analysis:** Yan Li.

**Funding acquisition:** Carlos F. Ibáñez, Meng Xie.

**Investigation:** Yan Li, Meng Xie.

**Resources:** Ming Tao.

**Supervision:** Carlos F. Ibáñez, Meng Xie.

**Writing – original draft:** Meng Xie.

**Writing – review & editing:** Carlos F. Ibáñez, Meng Xie.

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
