## [Editor Report · Decision Letter 0]

25 Jun 2025

Dear Dr Xie,

Thank you for submitting your manuscript entitled "Gpc3 Selectively Suppresses Subcutaneous Adipogenesis via Wnt Signaling in Diet-Induced Obesity" for consideration as a Research Article by PLOS Biology.

Your manuscript has now been evaluated by the PLOS Biology editorial staff as well as by an academic editor with relevant expertise and I am writing to let you know that we would like to send your submission out for external peer review.

Once your full submission is complete, your paper will undergo a series of checks in preparation for peer review. After your manuscript has passed the checks it will be sent out for review. To provide the metadata for your submission, please Login to Editorial Manager (https://www.editorialmanager.com/pbiology) within two working days, i.e. by Jun 27 2025 11:59PM.

Kind regards,

Luke

Lucas Smith, Ph.D.

Senior Editor

PLOS Biology

lsmith@plos.org

---

## [Decision Letter · Decision Letter 1]

2 Sep 2025

Dear Dr Xie,

Thank you for your patience while your manuscript "Gpc3 Selectively Suppresses Subcutaneous Adipogenesis via Wnt Signaling in Diet-Induced Obesity" was peer-reviewed at PLOS Biology. It has now been evaluated by the PLOS Biology editors, an Academic Editor with relevant expertise, and by several independent reviewers.

In light of the reviews, which you will find at the end of this email, we would like to invite you to revise the work to thoroughly address the reviewers' reports.

As you will see below, the reviewers find the study potentially interesting, but they have each raised a number of important and sometimes overlapping concerns which we think will need to be thoroughly addressed before we can consider your study for publication. While many of the reviewer suggestions are addressable with textual changes aimed at providing more detailed explanations and discussion, we do think that it will be important to experimentally strengthen the study by bolstering the mechanistic insights and testing the physiological consequences of Gpc3 mediated depot remodeling. We also think it will be critical to clarify recombination efficacy and cell type specificity of the knockout models.

Given the extent of revision needed, we cannot make a decision about publication until we have seen the revised manuscript and your response to the reviewers' comments. Your revised manuscript is likely to be sent for further evaluation by all or a subset of the reviewers.

**IMPORTANT - SUBMITTING YOUR REVISION**

*Re-submission Checklist*

*Published Peer Review*

*PLOS Data Policy*

*Blot and Gel Data Policy*

Sincerely,

Luke

Lucas Smith, Ph.D.

Senior Editor

PLOS Biology

lsmith@plos.org

REVIEWS:

Reviewer #1: In this study, Yan Li and colleagues investigate depot-specific regulation of adipogenesis in obesity, focusing on Glypican-3 (Gpc3) as a molecular determinant of adipose heterogeneity. By integrating human proteomics with mouse single-cell transcriptomics, they identify Gpc3 as an obesity-responsive gene showing reciprocal expression in subcutaneous (iWAT/ASAT) and visceral (eWAT/OVAT) depots in both species. Using ASPC-specific Gpc3 knockout models (constitutive and inducible), they demonstrate that Gpc3 loss selectively promotes high-fat diet-induced iWAT expansion, attributed to enhanced adipogenic differentiation and reduced proliferation of ASPCs. Mechanistically, they link this phenotype to downregulation of canonical Wnt signaling in iWAT progenitors, which can be rescued by pharmacological Wnt activation. The work positions Gpc3 as a potential target for modulating regional fat distribution in obesity. The study is interesting and potentially impactful; however, several substantive issues should be addressed before publication,

Major Points

1. The introduction is overly generic and does not sufficiently survey prior multi-omics studies of adipocyte progenitor cells (APCs) or highlight known differences between subcutaneous and visceral WAT progenitors. The novelty and specific knowledge gap addressed by this work are not clearly articulated, limiting the reader's appreciation of the study's significance.

2. The manuscript is difficult to follow in its current form. For example, Fig. 1 presents human ASAT/OVAT proteomics, while Fig. 2 shifts to murine scRNA-seq under various diet conditions without clearly linking the two datasets. The rationale for selecting Gpc3 as the focal molecule from these analyses is not convincingly presented. The connection between human whole-depot proteomics and mouse ASPC transcriptomics should be explicitly described.

3. Proteomic results (Figs. 1D-E, 4F) are presented in a high-throughput format but are sparsely interpreted. The statement that "depot-specific pathway signatures were maintained regardless of obesity status" requires further explanation. Similarly, scRNA-seq analysis (Fig. 2) does not fully exploit the dietary and depot-specific comparisons. The conclusion that "eWAT has a higher proportion of ASPC2, suggesting a propensity toward adipogenesis" is unsupported, as the relative adipogenic capacities of ASPC1 versus ASPC2 are not demonstrated in this study.

4. The claim that Gpc3 is the only HFD-responsive gene showing reciprocal expression between iWAT and eWAT should be supported by clear data. It is also important to clarify whether Gpc3 expression is predominantly localized to ASPCs in mice, as shown in human data (Fig. 3F), and whether its regulation differs between ASPC subsets. Including protein-level analyses in murine depots under different diets would strengthen this section.

5. In Fig. 5, Gpc3-deficient mice show increased iWAT mass and lineage tracing suggests enhanced adipogenesis. However, H&E staining in Fig. 4E shows larger adipocytes, suggestive of hypertrophy. The relationship between these findings should be reconciled. Moreover, the lineage tracing images (Figs. 5C, 5F) require improved quality and accurate quantification, as visual inspection and numerical data appear inconsistent. The higher GFP+ cell proportion in chow-fed mice compared to HFD-fed mice is unexpected and should be re-examined.

6. The observed differences in adipogenic capacity of preadipocytes from chow- versus HFD-fed mice (Fig. 6) are not clearly explained. Additional discussion of possible mechanisms would improve the reader's understanding.

7. The link between Gpc3 and Wnt signaling in regulating iWAT adipogenesis is based primarily on in vitro pharmacological modulation. In vivo genetic or targeted Wnt pathway manipulation in the Gpc3-deficient background would be needed to establish causality.

8. The study does not assess glucose tolerance, insulin sensitivity, or other systemic metabolic parameters, leaving the physiological implications of depot remodeling unresolved.

9. Both PdgfraCre and PdgfraCreERT lines target heterogeneous mesenchymal populations, and phenotypic differences between constitutive and inducible knockouts suggest possible developmental or indirect effects. Clarifying recombination efficiency and cell type specificity is essential.

Minor Points

1. Clarify statistical power and effect sizes for human proteomic comparisons.

2. Provide additional details on mouse cohort sex and age, as these factors influence depot expansion and progenitor biology.

3. Improve figure clarity when comparing reciprocal Gpc3 expression between human and mouse data by harmonizing presentation formats.

4. Consider adding a schematic summarizing the proposed depot-specific Gpc3-Wnt regulatory axis and its context-dependent effects.

Reviewer #2: This is an interesting study that identifies GPC3 as a novel obesity-responsive gene that has reciprocal effects in visceral and subcutaneous adipose tissue. The authors conduct a number of studies to examine its role in adipocyte expansion in both visceral and subcutaneous depots. Overall, most of these studies are strong, but I have a number of concerns. Notably, the authors describe Pdgfra-Cre as an ASPC-specific Cre, but it appears that Pdgfra has a broader expression pattern. Furthermore, they fail to establish the mechanism by which Gpc3 ablation increases adipogenesis. They do suggest that this occurs through Wnt signaling, but these are mechanistically weak and merely suggest that inhibiting Wnt signaling phenocopies the effect of Gpc3 knockout on Oil Red O staining.

Specific comments:

For all Two-way ANOVAs please report the p-values for both independent variables and the interaction. The way these are reported in the figures, I can't tell what factor the p-value is for. Also, please make it clear when reporting p-values for the Sidak multiple comparison test which values you are comparing.

Your title is "Gpc3 Selectively Suppresses Subcutaneous Adipogenesis via Wnt Signaling in Diet-Induced Obesity". You do show that deleting Gpc3 augments adipogenesis in subcutaneous adipose tissue, but I don't recall studies showing that adipogenesis can be suppressed by additional Gpc3. I feel that this title is slightly misleading and could be improved.

Figure 3. Why are you using heatmaps to show the difference in gene expression between 2 or 3 groups? This makes it incredibly hard to assess differences and variations within the group. Please replace these heatmaps with bar graphs like in 3C.

You report that "both fat depots exhibited reciprocal GPC3 expression patterns between lean and obese donors". It would be interesting to show this. Can you plot the expression of GPC3 against BMI for these donors? Also, it could be interesting to make similar graphs for iWAT and eWAT Gpc3 expression in mice, plotting its expression against body weight or total fat mass.

You use Pdgfra-Cre as an ASPC-specific Cre; however, it appears that this Cre is expressed in other cells/tissues. The lack of specificity needs to be clearly described in the manuscript.

Because Pdgfra-Cre causes a loss of Gpc3 in ASPCs, it ultimately causes a loss of Gpc3 in mature adipocytes. In Figure 4 when characterizing this mouse model, you need to show the knockout efficiency in both pre- and mature adipocyte populations from iWAT and eWAT. I would suggest that you isolate the SVF and mature adipocyte populations from iWAT and eWAT and assess the proportion of cells that have a genetic Gpc3 deletion in addition to the Gpc3 mRNA levels.

Figure 4E. The histology image is too low resolution to see. The bins for the bar graph seem a bit too big. The single bin that is 9,000 µm² ranging from 1,000 to 10,000 µm² encompassing ~80% is too wide to effectively describe the adipocyte cell sizes. I would suggest using bins every ~1,000 or 2,000 µm².

You mention that Gpc3 is located on the X chromosome in mice. Is it also located on the X chromosome in humans? Does sex affect the expression of Gpc3?

Some explanation of the mTmG reporter strain would be helpful to help the reader interpret these experiments.

Figure 5. Can you provide information about the size of the GFP+ cells? It appears that when you see increased numbers of GFP+ cells these cells do not grow as large. I wonder if given enough time these GFP+ cells would grow to the same size as those around them?

Figure 5 & 6. Your histology/ORO images are laid out differently between these two figures. In one you have the depot on the left and the genotype on the top. This is then reversed. It would be easier on the reader if you keep one style consistent throughout the manuscript.

Figure 6A. How did you control for the number of cells plated?

Figure 6D eWAT. The standard deviation in these bars is huge. I don't see how we can make any conclusions with this kind of variation.

Figure 6B,F. This could be improved if you also showed how these genes change in the eWAT.

Figure 6E. I don't see the eWAT in this figure.

Figure 7B: You say that "Wnt-related proteins were predominantly upregulated in mutant iWAT, while no consistent pattern was observed in eWAT." From the figure this appears to be reversed. Did you switch these two around? Also, do you have the numerical values for these heatmaps somewhere in this manuscript?

Figure 7C. You should also report the effect of CHIR99021 in Cre- cells. I'm not sure what the point is because it is known that inhibiting GSK3 will lead to increased β-catenin levels.

Figure 7E. You should also report the effect of XAV939 in Cre+ cells. I'm not sure what the point is because it is known that XAV939 promotes the degradation of β-catenin.

Figure 7D can be improved by including Gpc3flox + CHIR99021.

Figure 7F. While it is interesting that inhibiting Wnt signaling can increase ORO staining like Gpc3 knockout, this does not necessarily mean that Gpc3 is acting through Wnt signaling. More studies will need to be done to establish mechanism.

Mir-717 is within an intron of Gpc3 and is associated with obesity (PMID: 21152117). Is this affected in your model?

Reviewer #3: In the manuscript entitled "GPC3 selectively suppresses subcutaneous adipogenesis via Wnt signaling in diet-induced obesity", the authors investigate depot-specific responses of adipose tissue to dietary challenge by integrating proteomic profiling of human subcutaneous and visceral adipose tissue from obese and non-obese donors, along with scRNA-seq of mouse adipose stem and progenitor cells (ASPCs) from different depots under chow and high-fat diet conditions. They identify Glypican 3 (GPC3) as an obesity-responsive gene with reciprocal expression between adipose depots and generate ASPC-specific GPC3 knockout mice, showing an iWAT-specific hyperplasia phenotype mediated by canonical WNT signaling. This study addresses an important question and provides interesting observations; however, several points require clarification or additional validation to strengthen the manuscript.

Major and Specific Comments:

1) Figure 1: Is GPC3 protein enriched in ASAT in the obese group? Additionally, are there detectable differences in WNT signaling between subcutaneous and visceral adipose tissue under obese versus non-obese conditions?

2) Figure 2: The conclusion that "switching from CD to HFD induced a proportional shift from ASPC1 to ASPC2 at both 12- and 18-week time points, and an effect that was reversed upon switching back from HFD to CD" should be validated with flow cytometry or immunofluorescence staining. Similarly, the conclusion that "the proportion of Gpc3-expressing ASPCs increased under HFD and decreased upon reversion to CD" (Figure 3A-B) also requires experimental validation.

3) In Figure 3E, it would be better to compare GPC3 protein levels in ASAT and OVAT under obese and non-obese states. For Figure 3F, quantitative validation (e.g., GPC3+/PDGFRα+ cell fraction) should be provided to support the snRNA-seq results.

4) Figure 4: Does ASPC-specific Gpc3 deficiency alter the distribution or status of ASPC subpopulations? For example, does it promote expansion of the higher-adipogenic ASPC1 population?

5) Since Gpc3 is enriched in brown adipocyte progenitors, have the authors examined whether its deficiency affects thermogenesis and energy expenditure?

6) It would be informative to compare GPC3 levels in iWAT and eWAT under both chow and HFD conditions.

7) Figure 5C: The data suggests reduced adipogenesis in eWAT in the GPC3 knockout group. A more representative image would help clarify this. Additionally, the adipocyte size and quantification shown in Figure 5C seem inconsistent with Figure 4E; please provide an explanation.

8) Figure 5F: The data suggest higher adipogenesis rates in both iWAT and eWAT under chow diet compared with HFD conditions, even in control mice (relative to Figure 5C). This observation seems inconsistent with previous reports (e.g., PMID: 34562641). The authors should provide an explanation or discussion to reconcile this discrepancy.

9) Figure 7C: Assessing active β-catenin levels would strengthen the mechanistic link to WNT signaling.

10) The statistical analysis methods should be clearly indicated in each figure legend.

---

## [Decision Letter · Decision Letter 2]

14 Jan 2026

Dear Dr Xie,

Thank you for your patience while we considered your revised manuscript "Gpc3 Selectively Suppresses Subcutaneous Adipogenesis via Wnt Signaling in Diet-Induced Obesity" for publication as a Research Article at PLOS Biology. Your revised study has been evaluated by the PLOS Biology editors, the Academic Editor and the original reviewers.

As you will see in their comments, below, both reviewers 1 and 3 are fully satisfied by the revision. Reviewer 2 agrees that the revisions have strengthened the study, but s/he has a number of lingering concerns. Having discussed these last points, in detail, with the Academic Editor, our consensus opinion is the issues identified by reviewer 2 are important and that they will need to be thoroughly addressed in another revision before we can consider your study for publication.

One point, however, where our editorial opinion differs from reviewer 2, is the title. We are generally OK with you stating that "Gpc3 Selectively Suppresses Subcutaneous Adipogenesis", based on knockout data only. We will wait until we see your responses to reviewer 2, before weighing in on whether removing/toning down the 'via Wnt signaling' is warranted.

Given the extent of revision needed, which will likely require the generation of new data, we are providing another 3 month deadline for your revision. Please email us (plosbiology@plos.org) if you have any questions or concerns, or would like to request an extension. Please note, this will be the last round of experimental revision that we will allow.

We cannot make a decision about publication until we have seen the revised manuscript and your response to the reviewers' comments. Your revised manuscript is may be sent for further evaluation by the reviewers.

**IMPORTANT - SUBMITTING YOUR REVISION**

*Re-submission Checklist*

*Published Peer Review*

*PLOS Data Policy*

*Blot and Gel Data Policy*

Sincerely,

Luke

Lucas Smith, Ph.D.

Senior Editor

PLOS Biology

lsmith@plos.org

REVIEWS:

Reviewer #1: The authors addressed most of my concerns. I think this manuscript is suitable to be published.

Reviewer #2: Reviewer #2 Response: This is a revised article about Gpc3 suppression of subcutaneous adipogenesis via Wnt signaling. The authors conduct a number of studies showing depot specific effects of Gpc3 on adipocyte differentiation that greatly improves the manuscript, however I still have a number of significant concerns about these studies, especially about the mechanistic role of Wnt in their model.

Reviewer #2 Original Comment: This is an interesting study that identifies GPC3 as a novel obesity-responsive gene that has reciprocal effects in visceral and subcutaneous adipose tissue. The authors conduct a number of studies to examine its role in adipocyte expansion in both visceral and subcutaneous depots. Overall, most of these studies are strong, but I have a number of concerns. Notably, the authors describe Pdgfra-Cre as an ASPC-specific Cre, but it appears that Pdgfra has a broader expression pattern. Furthermore, they fail to establish the mechanism by which Gpc3 ablation increases adipogenesis. They do suggest that this occurs through Wnt signaling, but these are mechanistically weak and merely suggest that inhibiting Wnt signaling phenocopies the effect of Gpc3 knockout on Oil Red O staining.

Response: Please see below for detailed responses for all these comments.

Reviewer #2 Original Comment: For all Two-way ANOVAs please report the p-values for both independent variables and the interaction. The way these are reported in the figures, I can't tell what factor the p-value is for. Also, please make it clear when reporting p-values for the Sidak multiple comparison test which values you are comparing.

Response: Please see the revised legends for Figures 4-7, where we have now stated all p-values for the Two-way ANOVA interactions and specified the comparison groups for all Sidak post-hoc tests.

Reviewer #2 Response: I don't see the p-values for the genotype, treatment, and interaction terms. This is especially important for Figure 8 because these parts of the ANOVA tell you if the main effects are independent of each other.

Reviewer #2 Original Comment: Your title is "Gpc3 Selectively Suppresses Subcutaneous Adipogenesis via Wnt Signaling in Diet- Induced Obesity". You do show that deleting Gpc3 augments adipogenesis in subcutaneous adipose tissue, but I don't recall studies showing that adipogenesis can be suppressed by additional Gpc3. I feel that this title is slightly misleading and could be improved.

Response: While understanding the concern, we think that the title is reasonable and accurate based on the standard interpretation of mechanistic data in molecular biology. The reviewer is right that all our results are based on Gpc3 deletion. However, a common and accepted practice in functional genetics is that if the absence of a gene leads to an increase in a process, then the normal function of that gene is to suppress that same process. Therefore, our knockout data show that this augmentation occurs specifically when an endogenous suppressor (Gpc3) is removed, proving that suppression is its natural role.

Reviewer #2 Response: You have shown the necessity of Gpc3 in the suppression of adipogenesis, but have not shown the sufficiency. The title doesn't really reflect what you did in this paper - you showed that loss of Gpc3 enhances adipogenesis in iWAT but not eWAT. The title suggests that this study is about adding Gpc3 to suppress adipogenesis. Also, your new data have raised significant doubts about whether this occurs through Wnt.

Reviewer #2 Original Comment: Figure 3. Why are you using heatmaps to show the difference in gene expression between 2 or 3 groups? This makes it incredibly hard to assess differences and variations within the group. Please replace these heatmaps with bar graphs like in 3C.

Response: We have replaced those heat maps with bar graphs. Please see the revised Figure 3.

Reviewer #2 Response: Thank you.

Reviewer #2 Original Comment: You report that "both fat depots exhibited reciprocal GPC3 expression patterns between lean and obese donors". It would be interesting to show this. Can you plot the expression of GPC3 against BMI for these donors? Also, it could be interesting to make similar graphs for iWAT and eWAT Gpc3 expression in mice, plotting its expression against body weight or total fat mass.

Response: We have added these graphs as Figure 3G and H. Please see below for the panels and the related result description. "In addition, we found a positive correlation between GPC3 protein expression and BMI in the ASAT of the obese donor group (Fig. 3G). A similar positive correlation between Gpc3 mRNA level and body weight was seen in the iWAT of HFD-fed mice, whereas their eWAT exhibited a negative correlation (Fig. 3H)."

Reviewer #2 Response: In figure 3G&H, you have too few data points to really make conclusions about the slope of these relationships. I like being able to see the data presented like this and it is acceptable for human data where sample availability is limited, but I feel that the statements in the results section stating that a correlation was found are too strong. It is my opinion that the data should be shown, but avoid mentioning the findings of a correlation in the text because this indicates that you have done a robust study with enough data points to be confident in your conclusion.

Reviewer #2 Original Comment: You use Pdgfra-Cre as an ASPC-specific Cre; however, it appears that this Cre is expressed in other cells/tissues. The lack of specificity needs to be clearly described in the manuscript.

Response: We assessed Gpc3 mRNA levels in the pancreas and brain of the constitutive knockout models and found them to be reduced in the brain of mutant mice. The following texts and figure panels were added to the Results and Discussion sections:

Results "Of note, Gpc3 mRNA levels showed a decreasing trend in the brain, but not in the pancreas, of mutant mice (Fig. S3A). To determine if brain Gpc3 contributes to the phenotype, we generated Camk2aCre;Gpc3flox mice to delete Gpc3 specifically in forebrain neurons. On the same HFD regimen, these mice showed no differences in body weight or fat mass compared to Gpc3flox controls (Fig. S3B), indicating that brain Gpc3 does not play a major role in the observed obesity."

Discussion

"A notable finding in our study using the PdgfraCre driver was the observed reduction of Gpc3

expression not only in adipose tissue but also in the brain. This indicates that the PdgfraCre line, while effective in targeting adipocyte precursors, may also be active in progenitor cells common to other tissues or during early embryonic development. Thus, we cannot definitively rule out that the metabolic phenotype arises from a combined effect of Gpc3 loss across multiple organs. However, several lines of evidence suggest that adipose tissue is the critical site of action for Gpc3. First, the mutant's phenotype was characterized by a direct increase in iWAT adipogenesis, a logical consequence of disrupting a local signaling modulator. Second, forebrain-specific deletion of Gpc3 using Camk2aCre was insufficient to alter body weight or fat mass, excluding a major contributing role from the brain. Third, the mutant mice exhibited comparable glucose tolerance, insulin sensitivity, and energy expenditure to controls. This decoupling between the adipose phenotype and the absence of broader metabolic dysfunction suggests that the obesity originates from the adiposespecific loss of Gpc3, not from its concurrent loss in other organs."

Reviewer #2 Response: Thank you, however I do have some contention with your first point. The fact that adipogenesis is altered has no bearing on whether the mutation's effect was local. The mutation could be altering a hormone or neural signal affecting adipogenesis. Your lineage tracing and in vitro studies are better evidence of the cell autonomous effect than the argument you present here.

A minor point: your statement about Camk2a-Cre mice having a deletion in the forebrain neurons makes me ask the question if this is deleting Gpc3 from neurons involved in body weight regulation, such as food intake and thermogenesis. A statement about why this is a relevant model would be beneficial for the reader not already familiar with neuronal Cre lines.

Another minor point: referring to "mutant mice" is very confusing considering that you have multiple Cre lines. The jargon in this paper makes it harder to read.

Reviewer #2 Original Comment: Because Pdgfra-Cre causes a loss of Gpc3 in ASPCs, it ultimately causes a loss of Gpc3 in mature adipocytes. In Figure 4 when characterizing this mouse model, you need to show the knockout efficiency in both pre- and mature adipocyte populations from iWAT and eWAT. I would suggest that you isolate the SVF and mature adipocyte populations from iWAT and eWAT and assess the proportion of cells that have a genetic Gpc3 deletion in addition to the Gpc3 mRNA levels.

Response: We performed single-nucleus RNA sequencing on iWAT from control and mutant mice under HFD-feeding condition. Please see below the added texts and figure panels into the Results

section: "snRNA-seq of iWAT from HFD-fed mice confirmed the efficacy of the genetic model, revealing a 3-fold reduction in the proportion of cells expressing Gpc3 (predominantly ASPCs) in mutants compared to controls (Fig. 4I). While Gpc3 deletion did not drastically alter the proportional distribution of the two major ASPC subpopulations (Fig. 4J), it profoundly altered their transcriptional programs as reflected by a shift in the top 5 enriched pathways within both

populations (Fig. 4K). Specifically, ASPC1 transitioned from a thermogenic profile in controls to

an immunomodulatory state in mutants; whereas ASPC2 shifted its enrichment from extracellular matrix organization to pathways governing translational regulation (Fig. 4K)."

Reviewer #2 Response: The data in Fig. 4I very nicely show the cell type specific knockout of Gpc3 but, perhaps I missed it, I didn't see what depot these data are from. The mRNA in 4A,B suggests that the KO is not equally efficient in both depots so in whatever method you use to show KO efficiency will need to be done in both depots. I am concerned that your differences in the phenotypes of these two depots could be driven by fact that Gpc3 mRNA is almost totally removed from iWAT but only 50% removed from eWAT. If this turns out to be an unfortunate consequence of this particular Cre, the authors should be more cautious in their statements ascribing depot-specific effects to Gpc3. Furthermore, the analysis of the depot specific effects of Gpc3 is explored further in vitro using cell cultures derived from what I assume are these mice (ex. Fig. 6). Given the potential difference in knockout efficiency between depots, I am concerned that these in vitro studies could be comparing KO iWAT to nearly WT eWAT.

Reviewer #2 Original Comment: Figure 4E. The histology image is too low resolution to see. The bins for the bar graph seem a bit too big. The single bin that is 9,000 μm² ranging from 1,000 to 10,000 μm² encompassing ~80% is too wide to effectively describe the adipocyte cell sizes. I would suggest using bins every ~1,000 or 2,000 μm².

Response: We have increased the resolution of the histology image in Figure 4E for better

visualization. We have also changed the adipocyte size bins to every 2,000 μm².

Reviewer #2 Response: Thank you.

Reviewer #2 Original Comment: You mention that Gpc3 is located on the X chromosome in mice. Is it also located on the X chromosome in humans? Does sex affect the expression of Gpc3?

Response: We added the following texts and figure panels for this comment:

"Consistent with its location on the X chromosome, qPCR analysis revealed that Gpc3 mRNA

abundance was significantly higher in both iWAT and eWAT from wild-type female mice compared to their male counterparts at 4 months of age (Fig. S3C), suggesting that Gpc3 is a candidate escape gene from X-inactivation."

Reviewer #2 Response: Thank you. This is very interesting and could be a nice follow-up study. However, it leaves open a significant possibility that these findings are more sex specific. The authors should be careful to not indicate that these findings apply to males and females equally.

Reviewer #2 Original Comment: Some explanation of the mTmG reporter strain would be helpful to help the reader interpret these experiments.

Response: We added the following text for description of the mTmG reporter strain: "which labels Cre-expressing cells and their progeny with GFP while non-recombined cells express tdTomato (Muzumdar et al, 2007),"

Reviewer #2 Response: Thank you.

Reviewer #2 Original Comment: Figure 5. Can you provide information about the size of the GFP+ cells? It appears that when you see increased numbers of GFP+ cells these cells do not grow as large. I wonder if given enough time these GFP+ cells would grow to the same size as those around them?

Response: As quantified in Figure S4B, Gpc3-deficient (GFP+) adipocytes in the mutant iWAT were indeed significantly smaller than the control (Tomato+) adipocytes within the same tissue

microenvironment. While we cannot rule out the possibility of a potential developmental delay with absolute certainty, we would interpret the smaller size of the GFP+ adipocytes as a direct and persistent consequence of the cell-autonomous loss of Gpc3. The fact that GFP+ and Tomato+ adipocytes reside in the same microenvironment and exposed to the same systemic signals, yet maintain a size difference, suggests against a simple delay.

Reviewer #2 Response: Quantification of the size of the GFP+ cells is important because these are some of your strongest data supporting your arguments.

Reviewer #2 Original Comment: Figure 5 & 6. Your histology/ORO images are laid out differently between these two figures. In one you have the depot on the left and the genotype on the top. This is then reversed. It would be easier on the reader if you keep one style consistent throughout the manuscript.

Response: We have reorganized Figure 6 to improve clarity and presentation.

Reviewer #2 Response: Thank you.

Reviewer #2 Original Comment: Figure 6A. How did you control for the number of cells plated?

Response: We have added the following texts in the revised Methods section: "Prior to plating, viable cell numbers were quantified using a hemocytometer (Neubauer chamber) with trypan blue exclusion. Cells were then seeded in 12-well plates at 7.5 × 10⁴ cells/mL in a final volume of 2 mL per well."

Reviewer #2 Response: Getting accurate counts with a hemocytometer is very difficult. It appears that the authors have done well at plating the two genotypes at a similar density despite the challenges. However, a control to ensure that equal numbers WT and KO cells is very important to demonstrate the rigor of these studies (akin to the reason GAPDH is run on Wester Blots). The difference in BrdU incorporation in 6C already suggests that the cells may grow at different rates presenting a confounding, albeit interesting, factor in interpreting these data.

Reviewer #2 Original Comment: Figure 6D eWAT. The standard deviation in these bars is huge. I don't see how we can make any conclusions with this kind of variation.

Response: To ensure robustness, we re-ran the eWAT samples from the three independent experiments. This repeated analysis confirmed our initial finding, as reflected by the smaller

standard deviation now presented in Figure 6H. The figure panel is inserted below.

Reviewer #2 Response: Thank you. Also note that the bar in this figure appears to be a different shade of red.

Reviewer #2 Original Comment: Figure 6B,F. This could be improved if you also showed how these genes change in the eWAT. Figure 6E. I don't see the eWAT in this figure.

Response: We have now included data on eWAT in Figure 6 and have revised the Results section accordingly. The conclusion remains the same. We also added a paragraph in the Discussion section to address the unexpected observation in eWAT. Please see below for the revised texts.

Results

"To investigate the mechanisms underlying enhanced hyperplasia in Gpc3-deficient ASPCs, we

isolated ASPCs from iWAT and eWAT of HFD-fed control and mutant mice for in vitro primary culture. By days in vitro (DIV) 7, mutant iWAT-derived adipocytes exhibited significantly greater Oil Red O-stained lipid accumulation (Fig. 6A), along with increased expression of adipogenic (Pparg, Cebpa) and lipogenic (Dgat2) markers during differentiation (Fig. 6B). These changes were accompanied by reduced BrdU incorporation (Fig. 6C) and Ccnd1 expression (Fig. 6D), indicating impaired proliferative capacity. In contrast, eWAT-derived adipocytes from HFD-fed mice showed no differences in lipid accumulation (Fig. 6E) or proliferative capacity (Fig. 6G-H), despite elevated expression of adipogenic and lipogenic markers (Fig. 6F), consistent with the depot-specific regulatory mechanisms predicted by our proteomic analysis (Fig. 4F). Under CD condition, control and mutant iWAT-derived adipocytes showed comparable differentiation capacity (Fig. 6I) and similar Pparg, but not Cebpa or Dgat2 expression (Fig. 6J). Notably, mutant eWAT-derived adipocytes showed reduced differentiation capacity (Fig. 6K) with decreased expression of adipogenic and lipogenic markers (Fig. 6K). Together, these data demonstrate that Gpc3 deletion in ASPCs selectively enhances adipogenic potential while suppressing proliferation in iWAT under HFD condition."

Discussion

"Unexpectedly, in HFD-fed eWAT culture, adipogenic and lipogenic markers were upregulated in the mutant ASPC-derived adipocytes despite no detectable changes in differentiation or proliferative capacity were observed. In addition, in CD-fed eWAT culture, mutant progenitors exhibited reduced differentiation capacity but elevated expression of adipogenic and lipogenic markers. These uncoupling between gene expression and functional output suggests that eWAT ASPCs may exist in a transcriptionally primed state that does not readily translate into adipogenesis under obesogenic condition. Several mechanisms could account for this discrepancy, including depot-specific microenvironmental constraints such as inflammation and fibrosis, which are known to be more pronounced in eWAT and can suppress adipogenic execution even in the presence of elevated transcriptional programs (Kawai et al, 2021). Alternatively, post-transcriptional or epigenetic modifications may limit the translation of adipogenic regulators. Moreover, cell heterogeneity within the eWAT progenitor pool may result in a subset of cells driving marker expression without contributing to functional differentiation. Of note, the in vitro differentiation capacity of eWAT ASPCs is substantially lower than that of iWAT, which may further contribute to the observed discrepancies. Together, these findings highlight a diet- and depot-dependent complexity in Gpc3 function, in which elevated adipogenic transcriptional signatures in eWAT do not necessarily predict differentiation outcomes."

Reviewer #2 Response: Thank you.

Reviewer #2 Original Comment: Figure 7B: You say that "Wnt-related proteins were predominantly upregulated in mutant iWAT, while no consistent pattern was observed in eWAT." From the figure this appears to be reversed. Did you switch these two around? Also, do you have the numerical values for these heatmaps somewhere in this manuscript?

Response: We thank the reviewer for identifying this error. The figure has been corrected with the accurate labels. In addition, we included the numerical values for these proteins as Table S2.

Reviewer #2 Response: These graphs do not appear to correspond to the values in Table S2. Why are all of the data in 7B either completely red or completely blue? Are the Z-scores all exactly 0.6 or -0.6? Units for Table S2 are necessary.

Reviewer #2 Original Comment: Figure 7C. You should also report the effect of CHIR99021 in Cre- cells. I'm not sure what the point is because it is known that inhibiting GSK3 will lead to increased β-catenin levels.

Response: We have added the CHIR99021-treated Cre-cells in Figure 8F-G. Please see below for

the revised texts for this part: "Consistent with this, treatment with either recombinant Wnt3a protein (rWnt3a) (Fig. 8D-E) or CHIR99021 (a GSK3 inhibitor that stabilizes β-catenin) (Fig. 8F-G) recapitulated the effects of Axin1 knockdown, reversing the enhanced adipogenesis in mutant ASPCs. Notably, these Wnt activating treatments also reduced the adipogenic potential of control ASPCs, suggesting that canonical Wnt signaling is a universal inhibitor of adipogenesis, with Gpc3 serving as an upstream regulator of this pathway."

Reviewer #2 Response: I agree that these data suggest that canonical Wnt signaling is a universal inhibitor of adipogenesis, but I do not see evidence that Gpc3 is regulating Wnt/β-catenin.

Reviewer #2 Original Comment: Figure 7E. You should also report the effect of XAV939 in Cre+ cells. I'm not sure what the point is because it is known that XAV939 promotes the degradation of β-catenin.

Response: We have added the XAV939-treated Cre+ cells in Figure 8H-I. Please see below for the revised texts for this part: "Conversely, inhibition of Wnt signaling with XAV939 (a tankyrase inhibitor that promotes β-catenin degradation) in control ASPCs phenocopied the elevated adipogenesis observed in mutant cells (Fig. 8H-I versus 6A). Interestingly, XAV939 treatment could further enhance adipogenesis in mutant ASPCs, suggesting that the Gpc3 mutation, while sufficient to enhance differentiation, does not fully abolish the Wnt-mediated suppression of adipogenesis."

Reviewer #2 Response: I don't see evidence that Gpc3 loss reduces β-catenin. The fact that XAV939 treatment reduces β-catenin and increases differentiation to an apparently similar extent in both WT and cells with a loss of Gpc3 suggests that these effects on differentiation are independent of each other. If Gpc3 loss really lowered active β-catenin levels, then XAV939 wouldn't be able to reduce β-catenin much more and relatively would not have as great of an effect on differentiation.

Reviewer #2 Original Comment: Figure 7D can be improved by including Gpc3flox + CHIR99021.

Response: Please see response above for Figure 7C.

Reviewer #2 Original Comment: Figure 7F. While it is interesting that inhibiting Wnt signaling can increase ORO staining like Gpc3 knockout, this does not necessarily mean that Gpc3 is acting through Wnt signaling. More studies will need to be done to establish mechanism.

Response: We have incorporated additional experiments using both genetic (siRNA-mediated knockdown of Axin1) and ligand-based (recombinant Wnt3a protein treatment) approaches to modulate the Wnt pathway in primary iWAT ASPC cultures derived from HFD-fed control and Gpc3-deficient mice. The results from these complementary manipulations are consistent and further support our conclusions. The new results are presented in Figure 8A-G. Please see below for the revised texts for these results: "To determine whether the effects of Gpc3 on ASPC differentiation are Wnt-dependent, we knocked down Axin1, the gene that encodes the scaffold protein essential for β-catenin phosphorylation and degradation, in iWAT ASPCs from HFD-fed control and mutant mice. Successful Axin1 knockdown(Fig. 8A) increased active non-phospho β-catenin levels (Fig. 8B) and reversed the enhanced adipogenic potential of mutant ASPCs (Fig. 8C). Consistent with this, treatment with either recombinant Wnt3a protein (rWnt3a) (Fig. 8D-E) or CHIR99021 (a GSK3 inhibitor that stabilizes β-catenin) (Fig. 8F-G) recapitulated the effects of Axin1 knockdown, reversing the enhanced adipogenesis in mutant ASPCs. Notably, these Wnt-activating treatments also reduced the adipogenic potential of control ASPCs, suggesting that canonical Wnt signaling is a universal inhibitor of adipogenesis, with Gpc3 serving as an upstream regulator of this pathway. Conversely, inhibition of Wnt signaling with XAV939 (a tankyrase inhibitor that promotes β-catenin degradation) in control ASPCs phenocopied the elevated adipogenesis observed in mutant cells (Fig. 8H-I versus 6A). Interestingly, XAV939 treatment could further enhance adipogenesis in mutant ASPCs, suggesting that the Gpc3 mutation, while sufficient to enhance differentiation, does not fully abolish the Wnt-mediated suppression of adipogenesis. Collectively, these findings establish Gpc3 as a major, though not exclusive, regulator of the canonical Wnt pathway in ASPCs under HFD feeding condition."

Reviewer #2 Response: I think that the data in Figure 8 actually suggests that this phenomenon of Gpc3 affecting adipogenesis does not occur through Wnt signaling. It appears that whenever you increase β-catenin (Axin shRNA, rWnt3a, CHIR99021) differentiation is reduced while reducing β-catenin (XAV939) increases differentiation independent of Gpc3 genotype. Basically, Gpc3 genotype sets the baseline differentiation rate and modulating β-catenin adds or subtracts a consistent amount based on how much β-catenin is present. Also, I don't see any data suggesting that Gpc3 genotype affects active β-catenin levels. This is what I would expect the data to look like from a mutation that affects differentiation independently from β-catenin. Perhaps I missed it, but I do not see where the main effects of the ANOVA (Gpc3 genotype, treatment, interaction) are presented, and these would be useful in determining if the effects of β-catenin are independent of Gpc3.

I should have caught this in my first review, but are your Western blots for "active non-phospho β-catenin levels" for total β-catenin or an antibody specifically for the unphosphorylated β-catenin? (You should have a list of antibodies used that I may have missed). If the blots are for unphosphorylated β-catenin, total β-catenin should be used as the control.

Reviewer #2 Response New Minor Comments:

For all indirect calorimetry data, you need to show food intake and movement in addition to RER and energy expenditure. Perhaps I missed it, but I didn't see how you performed and analyzed these data. Also, what age were the mice and what diet were they on? Make sure you normalize the data by ANCOVA instead of to body weight as described in PMID: 40993210.

Figure 2D: Why are the colors apparently reversed in the fourth diagram?

On page 4 you mention "proteomic analysis of paired ASAT and OVAT…", from the text it is difficult to determine what kind of proteomic analysis you did without going to the methods. I think it would be easier to understand the experiment if you use more specific terms.

On page 6 you mentioned "control and mutant mice" without specifying which mutation you are talking about. I can figure it out from going to the figure, but readability would be improved if that was not necessary.

Figure 3F: How were GPC3 protein levels measured?

Figure 3 G-H legend: labeled as F, G, H.

For the GTT and ITT, how long were the mice on the diets?

Reviewer #3: The authors have addressed my questions, and I have no further comments.

---

## [Editor Report · Decision Letter 3]

13 Feb 2026

Dear Dr Xie,

Thank you for your patience while we considered your revised manuscript "Gpc3 Selectively Suppresses Subcutaneous Adipogenesis via Wnt Signaling in Diet-Induced Obesity" for publication as a Research Article at PLOS Biology. This revised version of your manuscript has been evaluated by the PLOS Biology editors and the Academic Editor.

The Academic Editor is satisfied by the changes made in response to the last round of review, and based his/her assessment of your revision we are likely to accept this manuscript for publication. However, before we can do so, we need you to address a few data and other policy-related requests in a last revision that we think will not take very long. These are detailed below.

**IMPORTANT - please address the following editorial requests:

1) TITLE - After some discussion within the team, we suggest that you change the title of your paper to:

"Gpc3 Selectively Suppresses Subcutaneous Adipogenesis in Diet-Induced Obesity"

^while we appreciate that your data indicate that Gpc3 calibrates the responsiveness of wnt signalling in this system, in the end we agree w/ reviewer 2 that that aspect of the title should be toned down.

2) ETHICS STATEMENT: Thank you for providing an ethics statement, in your methods section, for the human and animal studies performed here.

>For the human studies, please update this to include the approval number for your protocol and please indicate whether the study was conducted in accordance with the principles expressed in the Declaration of Helsinki.

>For the animal study please update the ethics statement to list the specific national or international regulations/guidelines to which your animal care and use protocol adhered. (examples of national guidelines include the 'NIH Guide for the Care and Use of Laboratory Animals' or the 'Guidelines for the ethical review of laboratory animal welfare People’s Republic of China National Standard GB/T 35892‐2018.')

3) DATA: Thank you for providing the underlying data for your study as a deposition to GEO and as supplemental tables.

>In the relevant section of our online system, please update your data availability statement to reference the data contained within the manuscript. For example, you can add the sentence "All other underlying data can be found within the manuscript file and supplementary material"

4) WESTERN BLOTS: Thanks also for providing the raw western blot data in a supplementary file. These images largely look good to me. However:

>>Please label each raw blot or gel image to clearly annotate the loading order and identity of experimental samples. Molecular weight markers should be included or indicated on the raw image, and any lanes not included in the final figure should be marked with an “X” above the lane label on the original blot/gel image. All labeling and annotation should be performed without obscuring any data or background bands.

5) CODE: Thank you also for providing the code related to your study on Github.

>Please note that we cannot accept sole deposition of code in GitHub, as this could be changed after publication. We therefore ask that you archive this version of your publicly available GitHub code to Zenodo. Once you do this, it will generate a DOI number, which you will need to provide in the Data Accessibility Statement (you are welcome to also provide the GitHub access information). See the process for doing this here: https://docs.github.com/en/repositories/archiving-a-github-repository/referencing-and-citing-content

We expect to receive your revised manuscript within two weeks.

*Published Peer Review History*

*Press*

Sincerely,

Luke

Lucas Smith, Ph.D.

Senior Editor

lsmith@plos.org

PLOS Biology

---

## [Editor Report · Decision Letter 4]

19 Feb 2026

Dear Dr Xie,

Thank you for the submission of your revised Research Article "Gpc3 Selectively Suppresses Subcutaneous Adipogenesis in Diet-Induced Obesity" for publication in PLOS Biology and thank you for addressing our editorial requests in this revision. On behalf of my colleagues and the Academic Editor, Young-Hwan Jo, I am pleased to say that we can in principle accept your manuscript for publication, provided you address any remaining formatting and reporting issues. These will be detailed in an email you should receive within 2-3 business days from our colleagues in the journal operations team; no action is required from you until then. Please note that we will not be able to formally accept your manuscript and schedule it for publication until you have completed any requested changes.

PRESS

We frequently collaborate with press offices. If your institution or institutions have a press office, please notify them about your upcoming paper at this point, to enable them to help maximize its impact. If the press office is planning to promote your findings, we would be grateful if they could coordinate with biologypress@plos.org. If you have previously opted in to the early version process, we ask that you notify us immediately of any press plans so that we may opt out on your behalf.

Sincerely,

Luke

Lucas Smith, Ph.D.

Senior Editor

PLOS Biology

lsmith@plos.org